# Antisense Oligonucleotide Therapy for the Nervous System: From Bench to Bedside with Emphasis on Pediatric Neurology

**DOI:** 10.3390/pharmaceutics14112389

**Published:** 2022-11-05

**Authors:** Man Amanat, Christina L. Nemeth, Amena Smith Fine, Doris G. Leung, Ali Fatemi

**Affiliations:** 1Moser Center for Leukodystrophies, Kennedy Krieger Institute, Baltimore, MD 21205, USA; 2Department of Neurology, Johns Hopkins University School of Medicine, Baltimore, MD 21205, USA; 3Center for Genetic Muscle Disorders, Kennedy Krieger Institute, Baltimore, MD 21205, USA

**Keywords:** antisense oligonucleotide, RNA therapy, spinal muscular atrophy, Duchenne muscular dystrophy, transthyretin amyloidosis, multiple sclerosis, Alexander disease, Canavan disease, Pelizaeus–Merzbacher disease, Angelman syndrome

## Abstract

Antisense oligonucleotides (ASOs) are disease-modifying agents affecting protein-coding and noncoding ribonucleic acids. Depending on the chemical modification and the location of hybridization, ASOs are able to reduce the level of toxic proteins, increase the level of functional protein, or modify the structure of impaired protein to improve function. There are multiple challenges in delivering ASOs to their site of action. Chemical modifications in the phosphodiester bond, nucleotide sugar, and nucleobase can increase structural thermodynamic stability and prevent ASO degradation. Furthermore, different particles, including viral vectors, conjugated peptides, conjugated antibodies, and nanocarriers, may improve ASO delivery. To date, six ASOs have been approved by the US Food and Drug Administration (FDA) in three neurological disorders: spinal muscular atrophy, Duchenne muscular dystrophy, and polyneuropathy caused by hereditary transthyretin amyloidosis. Ongoing preclinical and clinical studies are assessing the safety and efficacy of ASOs in multiple genetic and acquired neurological conditions. The current review provides an update on underlying mechanisms, design, chemical modifications, and delivery of ASOs. The administration of FDA-approved ASOs in neurological disorders is described, and current evidence on the safety and efficacy of ASOs in other neurological conditions, including pediatric neurological disorders, is reviewed.

## 1. Introduction

Human genetic modification and the alteration of gene expression have been developed to treat several disorders. The delivery of exogenous deoxyribonucleic acids (DNAs) to cells has been tested for years to restore expression of missing genes in the context of genetic disorders. Gene editing has also emerged to insert, delete, or replace genes in order to make specific changes to DNA. More recently, oligonucleotide-based therapy is being developed in an attempt to halt the progression of numerous disorders by affecting ribonucleic acid (RNA) or protein structure. Oligonucleotides are easy to synthesize and do not require integration into the genome, making their delivery less challenging. Different oligonucleotide structures have been designed and classified as (i) aptamers, which modify peptide and protein structures or enhance drug delivery; (ii) double-stranded RNA interference (RNAi), which reduces gene expression; (iii) double-stranded small activating RNAs (saRNAs), which induce gene expression; and (iv) antisense oligonucleotides (ASOs) (Figure 1) [1,2,3,4]. ASOs are single-stranded synthetic deoxynucleotide or ribonucleotide analogues, typically between 15 and 25 base pairs in length, that attach to complementary RNA sequences by Watson–Crick base pairing [4]. These chemically modified oligonucleotides are able to target both protein-coding (messenger RNA or mRNA) and noncoding RNA molecules (microRNA) to modulate gene expression. To date, nine ASOs have been approved by the US Food and Drug Administration (FDA), and a majority achieved marketing authorization in the last five years (Table 1). Several other ASOs have been developed and are currently in clinical trials to treat different neurological disorders, solid tumors (clinical trial identifiers NCT04862767, NCT04196257, NCT05267899, and NCT04504669), lymphoid malignancies (NCT04072458), leukemia (NCT02781883), hepatitis B virus infection (NCT05276297), moderate to severe COVID-19 (NCT04549922), myocardial infarction with reduced ejection fraction (NCT05350969), nonalcoholic steatohepatitis (NCT04483947), and end-stage renal disease (NCT05065463).

Oligonucleotides can decrease the levels of toxic protein (e.g., ASO, RNAi, and aptamer), increase the level of functional protein (e.g., ASO and saRNA), or alter the structure of a protein by mRNA splicing modification (e.g., ASO). Pegaptanib is the only FDA-approved aptamer to treat age-related macular degeneration by targeting vascular endothelial growth factor [1]. The current FDA-approved siRNAs include patisiran and vutrisiran for the treatment of polyneuropathy caused by hereditary transthyretin amyloidosis, givosiran for the treatment of acute hepatic porphyria, lumasiran for the treatment of primary hyperoxaluria type 1, and inclisiran for the treatment of heterozygous familial hypercholesterolemia or known cardiovascular disease [2,3,14,15,16]. Evidence is lacking to compare the safety and efficacy of different RNA therapeutic agents. Some studies have reported that siRNAs are more potent than ASOs in the in vitro setting due to the greater susceptibility of ASOs to nuclease degradation in cell culture [17,18,19]. One study, however, showed that ASOs and siRNAs have similar potency in cell culture [20].

This review provides an update on ASO mechanisms of action, design, modification strategies, and delivery methods to enhance efficacy. Finally, we discuss the uses and outcomes of ASOs in different neurological disorders.

## 2. Antisense Oligonucleotide Mechanisms

### 2.1. ASO Mechanisms of Action on Protein-Coding RNA

ASOs affect mRNA and protein expression using different mechanisms. Oligonucleotides can suppress protein expression via translational arrest due to the steric hindrance of ribosomal subunit binding (Figure 2A), inducing the RNase H endonuclease activity that cleaves RNA-DNA hybrids on mRNA (Figure 2B), or destabilizing pre-mRNA by inhibiting 5′ cap formation/modulating 3′ polyadenylation [21,22,23,24] (Figure 2C).

Other ASOs directly increase the production of a selected protein [25,26,27] (Figure 2D) by affecting upstream open reading frames (uORFs) within the 5′ untranslated region (5′ UTR) that inhibits the translational process in a manner proportional to the distance from the uORF to the start codon [28]. The uORFs are found in half of human mRNAs and typically reduce protein expression by 30% to 80%, with a modest impact on mRNA levels [29]. ASOs targeting uORFs were shown to increase protein synthesis by 30% to 150% in in vitro and in vivo settings in a dose-dependent manner [25]. Other translation inhibitory sequences in 5′ UTRs of mRNAs have been identified in mRNAs without uORFs, and ASOs targeting these regions were found to increase protein synthesis up to 2.7-fold in different cell types [26]. Another mechanism of increasing gene expression is to target nonproductive alternative splicing in pre-mRNAs. These alternative splicing regions produce nonfunctional mRNAs containing premature termination codons and should be degraded by nonsense-mediated decay. This pathway can also reduce the production of other productive mRNA transcripts [30]. Silencing nonproductive alternative splicing should, therefore, reduce the activity of nonsense-mediated mRNA decay with subsequent increase in the levels of productive mRNA transcripts. A recent ASO with a similar mechanism has been developed and tested in a mouse model of Dravet syndrome and was shown to increase productive *SCN1A* mRNA transcripts sixfold and the voltage-gated sodium channel α subunit Na_v_.1.1 by over 50% in a dose dependent manner [27]. An ongoing phase I/II clinical trial aims to assess the safety and efficacy of this drug in different doses (NCT04442295).

ASOs are also used to modify pre-mRNA splicing. Alternative splicing is critical in allowing a single gene to encode distinct proteins, and pathological changes in this process can be the cause or consequence of different disorders, including malignancies, Alzheimer’s disease, and familial dysautonomia [31,32,33]. Different RNA binding proteins regulate the splicing process by attaching to specific nucleotide sequences. ASOs that are complementary to pre-mRNA splicing enhancers or splicing silencers can cause blocking reactions at binding sites to induce skipping or inclusion of exons; respectively [34,35]. A few ASOs with noncomplementary “tails” have also been tested to determine if they can alter mRNA splicing without blocking reactions. The noncomplementary sequence in these ASOs facilitates exon splicing by forming a splicing enhancer (Figure 2E) [36,37].

### 2.2. ASO Mechanisms of Action on Noncoding RNA

MicroRNAs (miRNAs) are small molecules consisting of 18 to 25 nucleotides that regulate gene expression at the posttranscriptional level [38]. Several miRNAs, such as miRNA-21 and miRNA-155, have been linked to the progression of different malignancies [39,40]. These two miRNAs are oncogenic molecules that prevent the expression of phosphatases and proteins related to mitotic regulation [41,42]. Their expressions are associated with solid tumors and lymphomas. Knockdown of these miRNAs using ASOs inhibits unregulated cell proliferation and enhances programmed cell death (Figure 2F). Some nonmalignant conditions such as hepatitis C (HCV), Alport syndrome, and cardiac remodeling after myocardial infarction were also found to be correlated with atypical activation of miRNAs that may be inhibited by ASOs [43,44,45].

Other types of noncoding RNAs can also be the targets of ASOs. Three ongoing phase I clinical trials are assessing the safety of ASOs targeting a noncoding transcript (*UBE3A-ATS*) to activate paternal *UBE3A* gene expression in individuals with Angelman syndrome (NCT04259281, NCT04428281, and NCT05127226) (Figure 2G).

## 3. Design Considerations of Antisense Oligonucleotides

ASOs require a stable structure and high-affinity target binding to provide effective and safe treatment. The main parameters that should be considered in ASO design include the following.

### 3.1. Accessibility of the Target Sequence That Binds the ASO

Prediction of RNA secondary structures is an important factor in assessing accessible regions for ASO binding. Although the target sequence needs to be partly accessible, the length of the target region does not appear to affect efficacy [46]. While evidence on the most accurate predictive algorithm is lacking, *mfold* (http://www.unafold.org/mfold/applications/rna-folding-form.php (accessed on 14 September 2022)) is the most widely used open-access program that predicts optimal and suboptimal RNA secondary structures based on their thermodynamic features [47]. The software also reports “ss-count,” which is defined as the number of times that a nucleotide is single-stranded in a group of secondary RNA structures [47]. *RNAfold* (http://rna.tbi.univie.ac.at/cgi-bin/RNAWebSuite/RNAfold.cgi (accessed on 14 September 2022)) is another common and freely accessible program that predicts RNA secondary structures and reports minimum free energy according to a loop-based energy model and a dynamic programming algorithm [48]. Other programs with predictive algorithms for RNA secondary structures include *RNAstructure* (https://rna.urmc.rochester.edu/RNAstructure.html (accessed on 14 September 2022)), *SPOT-RNA* (https://sparks-lab.org/server/spot-rna (accessed on 14 September 2022)), and *RNAsoft* (http://www.rnasoft.ca (accessed on 14 September 2022)) [49,50,51].

### 3.2. Stability of the ASO-RNA Duplex

The GC content and basic melting temperature (T_m_) of oligonucleotides, as well as the ASO-RNA binding energy (∆G^o^_37_), should be assessed to design an ASO capable of a stable attachment to the complementary sequence [46,52]. An ASO GC content of approximately 60% is ideal for providing a thermodynamically stable ASO-RNA duplex [53]. Based on the analysis of sequence motifs in ASOs, the presence of four consecutive Gs or “CCGG” motifs were, however, found to reduce antisense activity [54]. Furthermore, ASOs including “ACTG,” “AAA,” and “TAA” motifs are associated with lower activity, while the presence of “CCAC,” “TCCC,” “ACTC,” “GCCA,” and “CTCT” motifs are associated with higher antisense activity [54]. Finally, basic T_m_ and ASO-RNA binding energy are significantly higher among ASOs with therapeutic efficacy, compared to ineffective ones [46,52]. The basic T_m_ ranges from 36 °C to 63 °C among effective ASOs. The ASO-RNA binding energy should be ∆G^o^_37_ ≥ −8 kcal/mol to provide a more effective response [46,52]. The above-mentioned oligonucleotide properties can be measured using the online software Oligo Calc (http://biotools.nubic.northwestern.edu/OligoCalc.html (accessed on 16 September 2022)) [55].

### 3.3. Specificity of the ASO Target Sequence

Aspecific and unintended genetic modifications are possible, as off-target mRNAs may have a partial sequence overlap with the intended ASO complementary target [25]. NCBI’s BLAST server (https://blast.ncbi.nlm.nih.gov (accessed on 16 September 2022)) is a useful tool to identify mRNAs with partial or complete nucleotide overlap with the ASO target sequence [56]. As a follow-up, the accessibility and binding energy of the off-target sequence should be evaluated to ensure limited interference. Further steps can include investigation of the effect of the ASO on off-target protein production in an in vitro or in vivo setting.

## 4. Antisense Oligonucleotide Modifications

There are several challenges in delivering ASOs to their sites of action. ASOs need to be imported into the cells and may require passage through the bloodstream and biological barriers (depending on the route of administration). Typically, diffusion into cells is limited by the negative charge of ASOs. Once in the cells, ASOs must evade lysosomal enzymes (e.g., nucleases) to avoid degradation, and secretory vesicles and endosomes, which can inhibit their export. To help overcome these challenges, several chemical modifications to nucleic acid subunits, including phosphodiester bond, five-carbon sugar, and nucleobase, have been suggested, which may improve ASO stability and function and reduce off-target toxicity (Figure 3). Unmodified oligonucleotides are 5- to 10-fold more susceptible to degradation by endo- and exonucleases compared to modified oligonucleotides.

### 4.1. Modified Phosphodiester Bond

The replacement of nonbridging oxygens in the phosphate groups of nucleotides by a sulfur atom forms a phosphorothioate (PS) backbone that is known as the first-generation ASO [57,58]. The first FDA-approved ASO, fomivirsen, has 21 nucleotides and only includes PS modifications. PS-ASOs are resistant to nucleases, and they increase cellular uptake and bioavailability twofold compared to unmodified ASOs [59]. In addition, they bind plasma proteins, such as albumin, that reduce renal excretion and lead to a longer half-life compared to unmodified ASOs [60]. This modification, however, decreases the oligonucleotide melting temperature and affinity to its target. Furthermore, the interaction of PS-ASOs with proteins elicits an immune reaction, mainly by complement activation, and PS-ASOs containing CpG dinucleotides activate Toll-like receptors [61]. Immune stimulation is shown to cause aspecific adverse events in in vivo and clinical settings such as serum transaminase activation, thrombocytopenia, and activated partial thrombin time prolongation [62].

Other modified phosphodiester bonds include the replacement of nonbridging oxygen in the phosphate groups with boron (boranophosphate), methyl (methylphosphonate), and nitrogen (phosphoramidite). Insertion of an ethyl group to the nonbridging oxygen forms a P-ethoxy backbone. ASOs with PS or boranophosphate backbones recruit RNase H to degrade mRNA, but methylphosphonate, phosphoramidite, and P-ethoxy backbones do not support RNase H activity [57,63,64,65]. Studies showed that phosphoramidite ASOs had a higher binding affinity, nuclease stability, and specificity of action compared to PS-ASOs [66]. Data from in vivo studies also reported that phosphoramidite ASOs are more potent in the treatment of malignancies, with faster renal clearance compared to PS-ASOs [67,68].

### 4.2. Modified 2′ Nucleotide Sugar

The 2′-O-methyl (2′-O-Me), 2′-O-methoxyethyl (2′-O-MOE), methylene 2′-O-4′-C linkage, constrained ethyl, and 2′-fluoro (2′ F) are common modifications in oligonucleotides. Each alteration at the 2′ position increases the melting temperature, improving the nuclease resistance, half-life, and target affinity compared to PS-ASOs [69]. Mipomersen (2′-O-MOE-PS), nusinersen (2′-O-MOE-PS), and inotersen (2′-O-MOE-PS) are the current FDA-approved ASOs with 2′ ribose modifications.

Locked nucleic acids (LNA) contain a 2′-O-CH2-4′ bridge in each ribose that improves thermodynamic stability and mRNA target binding by enhancing the lipophilic nature and forming rigid conformation [70]. LNAs also affect double-stranded DNA and form duplex and triplex structures to regulate the transcription of selective genes [71]. ASOs containing 2′, 4′-constrained ethyl-modified (cEt) residues provide high mRNA binding affinity and in vivo potency [72]. Although there are no FDA-approved LNAs or cEts, several preclinical studies reported the safety and efficacy of these modified oligonucleotides in the treatment of osteoarthritis, Huntington’s disease, facioscapulohumeral muscular dystrophy (FSHD), myotonic dystrophy type 1, and cystic fibrosis [72,73,74,75,76]. A few LNAs have been designed to inhibit expression of miRNAs related to T-cell lymphoma (anti-miRNA-155 or cobomarsen), HCV infection (anti-miRNA-122 or miravirsen), Alport syndrome (anti-miRNA-21 or lademirsen), and cardiac remodeling after myocardial infarction (anti-miRNA-132 or CDR132L) [44,45,77,78,79]; however, the recent trial on cobomarsen was terminated early for reasons unrelated to drug efficacy or safety (NCT03713320) and miravirsen and lademirsen (NCT02855268) have been discontinued due to the limited efficacy [80]. The safety and efficacy of CDR132L (NCT05350969) is being investigated in ongoing double-blind placebo-controlled phase II clinical trials.

The 2′-deoxy-2′-fluoro-arabinonucleic acid (FANA) ASO mimics DNA secondary structure and is the only ASO with a 2′ sugar modification that supports RNase H activity. The RNA cleavage rate when using PS-FANA is, however, slower than that when using PS-DNA structure [81]. FANA-ASOs have been successfully used in different in vitro and in vivo studies to target human immunodeficiency virus (HIV), Crohn’s disease, asthma, cancer-related androgen receptor–RNA complexes, angiogenesis, and inflammation due to spinal cord injury [82,83,84,85,86,87].

ASOs with modifications in the 2′ position of a nucleotide sugar can be used to reduce or increase protein translation or alter splicing. Although most of these oligonucleotides are not a substrate of RNase H, a chimeric ASO/DNA that consists of an ASO strand embedded within a single-stranded DNA template has been shown to be effective in binding target mRNA and activating RNase H with less immunogenicity and off-target toxicity compared to PS-ASO. These RNase H-inducing structures are referred to as “gapmers,” due to their central DNA gap flanked by oligonucleotides with chemical modifications. The DNA gap restores RNase H-mediated cleavage of mRNA, and the modified nucleotides resist nuclease cleavage of ASOs [52]. The efficacy of gapmers relies on the size of the central gap and the type of 2′ modification in the wings. Gapmers with modified 2′-O-MOE provide more consistent reduction in mRNA levels, compared to modified 2′-O-Me. The LNA/DNA chimeric ASO with seven to ten DNA nucleotides flanked by two to three LNA oligomers on both ends is effective in activating the RNase H enzyme. Studies on FANA-ASOs showed that repetitive pairs of “modified FANA–unmodified DNA” were more potent than FANA-gapmers [88].

### 4.3. Modified Phosphodiester Bond and Five-Carbon Sugar

Phosphorodiamidate morpholino oligomers (PMO) and peptide nucleic acids (PNA) were designed with further chemical alterations to enhance both thermodynamic stability and nuclease resistance. They are charge-neutral oligonucleotides with high mRNA target affinity and biostability. They have, however, fast renal clearance due to minimal interactions with proteins [89]. PMOs are formed by the substitution of nucleotide sugars with a six-membered morpholine subunit. In addition, the phosphodiester bond is replaced by the phosphorodiamidate linkage. The morpholine ring increases water solubility and the lack of a carbonyl group prevents PMO cleavage by proteases and esterases [89]. PMOs are not substrates of RNase H; they have been designed for splicing modification or translational arrest by steric hindrance of ribosome assembly. The cellular uptake of PMO in an in vitro setting is low but conjugated short cell-penetrating peptides, Endo-Porter, Lipofectamine, liposomes, and nucleofection facilitate cellular uptake [90,91,92]. Eteplirsen, golodirsen, viltolarsen, and casimersen are the current FDA-approved PMOs.

The replacement of nucleotide sugars and phosphates by the pseudo-peptide skeleton characterizes the PNAs. The chemical modifications include N-2-aminoethyl glycine units with a flexible methyl carbonyl linker attached to the nucleobase. PNAs are resistant to nucleases and peptidases that cause high stability in biological fluids. High binding affinity, as well as high sensitivity and specificity, are other PNA features. These oligonucleotides do not support RNase H activity, but they can inhibit the translational process by steric hindrance of ribosomal assembly [93]. Furthermore, PNAs can attach to double-stranded DNA and form several conformations that establish transcriptional arrest [94,95]. Low water solubility and reduced cellular uptake are the current challenges that decrease PNA efficacy. The CPP-conjugated PNA is a PNA variant developed to improve PNA cellular uptake [96,97].

### 4.4. Modified Nucleobase

Nucleobase modification is less common than phosphodiester bond or sugar modification. The chemical alteration in nucleobase positions should not interfere with base pairing. The 5-methyl substitution on pyrimidine nucleobases is the most common modification and is reported to increase ASO melting temperature and binding affinity while incurring less immune stimulation compared to the PS backbone [98]. Furthermore, it enhances target sequence selectivity and reduces off-target toxicity. Mipomersen (5-methylcytidine and 5-methyluridine) and inotersen (5-methylcytidine) are the two FDA-approved ASOs with nucleobase modifications. G-clamp is another pyrimidine alteration that includes a cytosine analogue containing phenoxazine residues to provide five hydrogen bonds with a complementary guanosine base on target [99,100]. Studies on PNA showed that a single G-clamp increased the melting temperature of the PNA-target duplex by 23 °C and enhanced the PNA potency [99,100].

## 5. Antisense Oligonucleotide Delivery

Although chemical modifications improve ASO delivery by preventing nuclease cleavage and increasing binding affinity, other strategies have been developed to enhance cellular uptake and drug internalization to further improve delivery and maximize therapeutic potency. Nucleofection and commercial transfection reagents, including Lipofectamine, Lipofectin, Fugene, and Endo-Porter, are commonly used in in vitro settings to deliver oligonucleotides into cells [91,101]. Nucleofection can enhance cellular uptake of oligonucleotides by applying an electric field that forms transient pores in the cell membrane. The abovementioned transfection reagents mostly form a complex with oligonucleotides in cell media that is lipophilic, increasing the ability of ASO to cross cell membranes and escape endosomes. Oligonucleotides are then released from the complex and bind to the complementary target. Viral vectors, conjugated peptides, antibodies, and other ligands, as well as nanoparticles (NPs) and extracellular vesicles (EVs), are possible means of advancing ASO delivery in preclinical and clinical studies (Figure 4). None of the current FDA-approved ASOs have delivery particles and all are administered to patients as naked oligonucleotides. Some clinical studies are currently assessing the efficacy of ASOs with delivery particles, including eplontersen (NCT05071300), BP1001 (NCT02781883), vesleteplirsen (NCT04004065), DYNE-101 (NCT05481879), and DYNE-251 (NCT05524883). Phase I studies showed that eplontersen and BP1001 were safe [102,103]. Eplontersen is a ligand conjugated antisense drug developed for the treatment of hereditary transthyretin amyloidosis, and BP1001 is an ASO loaded on an NP to treat leukemia.

### 5.1. Viral Vectors

For decades, recombinant viruses have been engineered that contain no pathogenic or replicating features but can cross cell and nuclear membranes to deliver nucleic acids into the nucleus of cells. Retrovirus, lentivirus, adenovirus, and adenoassociated virus (AAV) are widely used in gene therapy, with each vector having specific advantages and disadvantages for use. Retroviral vectors integrate into the human genome, increasing the risk of cancer (because of insertional mutagenesis) or the alteration of gene activity close to the insertion site [104]. Lentiviruses can also integrate into the genome, and recent clinical studies of sickle cell disease and X-linked adrenoleukodystrophy reported development of myelodysplastic syndrome in a few participants one to five years after lentiviral gene therapy [105,106]. Adenovirus and AAV are not found to increase cancer risk, but adenoviral vectors have high immunogenicity and may induce inflammation of transduced tissues [107,108]. Low packaging capacity is the major limitation of AAV vector [109]; however, nonintegrating lentiviruses and high-capacity AAV have been developed in an attempt to resolve the current risks and limitations of viral vectors. These vectors need to be tested to determine both safety and efficacy [110,111].

One of the earliest ASO delivery methods involved preparing viral vectors that included target cDNA with antisense direction [112,113]. The cDNA was then transduced into the cell nucleus and subsequent mRNA transcripts bound to the target sequence to silence the translational process. This protocol was successfully used in in vivo settings to knockdown angiotensin II receptor type I to treat hypertension [112,113]. Despite some success, the large size of the antisense mRNA is a major disadvantage of this method and may affect off-target mRNAs, leading to different side effects. Further studies have introduced viral vectors with modified small nuclear RNAs (snRNAs) to cells in an attempt to alter exon splicing in pre-mRNAs. The uridine-rich (U7) snRNA is a component of U7 small nucleotide ribonucleoprotein (U7 snRNP) and includes an antisense sequence designed to recognize histone pre-mRNAs and process their 3′ end [114]. The utility of this technique can be expanded, and this antisense sequence can be replaced by custom nucleotides to target other pre-mRNA sequences. The *SMN2*-antisense U7 snRNA was successfully used with lentivirus and adenovirus to increase exon 7 inclusion and restore functional SMN protein in spinal muscular atrophy (SMA) [115,116]. Another study assessed the efficacy of 11 different *DMD*-antisense U7 snRNAs using lentivirus and AAV in in vitro and in vivo settings, respectively [117], and reported that different U7 snRNA constructs could be combined into a single AAV vector to achieve skipping of multiple exons and potentially restore functional DMD protein for Duchenne muscular dystrophy (DMD).

### 5.2. Conjugated Peptides

Cell-penetrating peptides (CPPs) have fewer than 30 amino acids and were first discovered in 1988 through examination of the function of short arginine-rich peptide encoded by HIV-1 or Tat peptide to carry proteins and cross cell membranes to deliver cargo to cytosol [118,119]. CPPs facilitate the transport of different cargo molecules that are not able to cross cell membranes, including oligonucleotides and proteins. The underlying mechanisms of CPP internalization are not completely understood; however, several studies suggest that CPPs enter cells by directly penetrating the cell membrane or by endocytosis [120,121,122,123,124]. Cell membrane penetration is known as membrane transduction and can be achieved by the interaction of the negatively charged phospholipid layer with the positively charged CPPs that destabilize cell membranes or form transient structures including pores and inverted micelles [123,124] (Figure 5A). Endocytic internalization can be directed toward macropinocytosis, clathrin-mediated, caveolin-mediated, or clathrin/caveolin-independent pathways [125,126,127,128]. The subsequent release of peptides and ASO cargo from vesicles is the main limiting factor that determines the effectiveness of drug delivery after endocytosis (Figure 5B). An ongoing clinical trial is assessing the safety and efficacy of peptide-conjugated eteplirsen (vesleteplirsen) in patients with DMD (NCT04004065).

### 5.3. Conjugated Antibodies

Antibodies conjugated with ASOs are able to direct oligonucleotides to specific tissues and increase cellular uptake. Prior endeavors to assess the efficacy of antibody–ASO conjugates were performed on glioblastoma cell lines [129]. Transferrin receptors are highly expressed in brain cancers and conjugation of ASOs with anti-transferrin receptor monoclonal antibodies showed a threefold increase in cellular uptake compared to naked ASOs and twofold higher than ASOs conjugated with an aspecific IgG antibody [129]. A recent study also showed that ASOs conjugated with anti-CD44 and anti-EphA2 monoclonal antibodies were effective in reducing DRR/FAM107A expression in glioblastoma stem cells [130]. A preclinical study on precursor B-cell acute lymphoblastic leukemia reported that an anti-CD22 antibody-MXD3 ASO conjugate decreased MXD3 expression by 60% to 70% in different cell lines compared to untreated cells. The leukemia mouse model treated with an anti-CD22 antibody-MXD3 ASO conjugate survived significantly longer than untreated mice [131]. Two ongoing clinical trials are evaluating the safety and efficacy of antibody-conjugated ASOs in patients with myotonic dystrophy type I (NCT05481879) and DMD (NCT05524883).

### 5.4. Other Conjugated Ligands

High binding affinity and selectivity to their targets are major properties of aptamers that make them suitable carriers of therapeutic drugs. ASOs can attach to aptamers by covalent bonds or Watson–Crick base pairing [132]. The hydrophobic alteration of aptamer nucleobases is found to enhance cellular uptake and internalization of ASOs by improving hydrophobic interactions between aptamers and membrane proteins [133].

A few other ligands have been developed and conjugated with ASOs to improve target specificity. To increase ASO uptake by liver cells, acetylgalactosamine (GalNAc) has been attached to ASOs, as GaINAc has a high binding affinity with the asialoglycoprotein receptor located on hepatocytes [134]. A recent preclinical study successfully administered ASOs conjugated with a nonselective monoamine transporter inhibitor (indatraline) to mice in an attempt to reduce alpha-synuclein expression in brain-stem monoamine nuclei [135].

### 5.5. Nanoparticles

One of the main goals of nanomedicine is to deliver therapeutic drugs to tissues in a controlled manner. Drugs can be loaded on NPs by covalent bonds or noncovalent interactions such as hydrophobic, electrostatic, hydrogen bonding, or steric immobilization [136,137,138,139,140]. Oligonucleotides carried by NPs can easily cross cell membranes, are at lower risk of nuclease cleavage and have higher bioavailability [141]. Several organic NPs, including polymeric NPs (e.g., poly (d, L-lactide-co-glycolide) or PLGA, poly(ethylene glycol) or PEG, poly (ethyleneimine) or PEI, and polyamidoamine dendrimer)), lipid-based NPs (e.g., polymersomes, solid lipid NPs, liposomes, and micelles), lipid–polymer hybrid NPs, and exosomes, have been commonly used for ASO delivery, with promising results (Figure 4C).

To target specific tissues, polymeric NPs should be conjugated with ligands. Additionally, in vivo studies showed that polymeric NPs were not able to cross the blood–brain barrier (BBB) when they were systematically administered and required surface modification to overcome this issue [142]. A recent study designed glucose-coated polymeric NPs attached to ASOs to bind glucose transporter 1 (GLUT1) receptor of brain capillary endothelial cells and reported efficient accumulation of ASOs in the brains of mice one hour after intravenous (IV) administration of drug [143]. Several limitations have been reported for cationic-charged lipid-based NPs related to their high permanent positive charge that aggregates with anionic structures. They are found to activate immune cells and complement, which induces over-production of proinflammatory cytokines and subsequent cytotoxicity with aspecific adverse events [142,144]. In addition, these NPs have a low half-life due to fast renal clearance [142,144,145]. To address these limitations, ionizable cationic lipids were developed. A recent phase I clinical trial tested one such lipid, BP1001, an ASO that includes anti-growth factor receptor-bound protein 2 (Grb2) oligonucleotides with P-ethoxy backbone loaded on dioleoylphosphatidylcholine (DOPC), an ionizable cationic lipid, in individuals with acute myeloid leukemia, Philadelphia-chromosome-positive chronic myeloid leukemia, acute lymphoblastic leukemia, or myelodysplastic syndrome. This drug was reported to be safe with beneficial effects [103]. Lipid–polymer hybrid NPs contain a polymeric core coated by lipid layer. Polymeric cores include the therapeutic agent and other drug-delivery particles (e.g., peptide, aptamer, PEG, antibodies, and lipophilic derivatives) may be added to the lipid layer to further enhance delivery [146,147,148]. A recent preclinical study on glioblastoma cells found that about 80% of lipid–polymer hybrid NPs containing anti-miRNA-21 antisense oligonucleotide were delivered into the cells [149]. The main limitation of lipid–polymer hybrid NPs is their quick elimination from systemic circulation due to fast renal clearance [146].

Calcium phosphate, silica, iron oxide, and gold NPs are inorganic NPs used with ASOs in a few studies [150,151,152,153]. Inorganic NPs have higher thermodynamic stability and are more resistant to degradation, compared to organic NPs [154].

### 5.6. Extracellular Vesicles

EVs are biovesicles secreted from almost all types of living cells into extracellular fluid and play a critical role in intercellular communication by shuttling proteins, lipids, and nucleic acids from donor to acceptor cells [155]. Different types of EVs have been identified. The main types include exosomes, microvesicles, oncosomes, and apoptotic bodies [156]. A preclinical study reported that ASOs loaded into the red blood cell-derived EVs, including exosome-like and microvesicle-like particles, reduced tumor size in breast cancer and suppressed acute myeloid leukemia progression [157]. Most studies on ASO delivery with EVs have focused on the safety and efficacy of exosomes, while evidence on other EVs is limited.

Exosomes include several subtypes, and the cellular uptake of each subtype is highly specific. Codelivery of exosomes and ASOs may, therefore, enhance ASO delivery into cells and reduce off-target toxicity. Recent in vitro and in vivo studies found that ASOs encapsulated by exosomes were safe and effective in the treatment of glioblastoma, Parkinson’s disease, and liver fibrosis [158,159,160]. A simple coincubation of oligonucleotides with exosomes can be an efficient way to improve drug delivery into the cells [161,162]. Exon-skipping ASOs loaded into the exosomes using coincubation increased the levels of dystrophin by 18-fold in a DMD mouse model compared to naked ASOs [161]. Our knowledge of endosomal characteristics, physiology, and internalization of exosomes is limited, and providing pure and specific exosomes for use is currently challenging [163].

## 6. Clinical Trials Using ASOs in Neurological Disorders

The current FDA-approved ASOs in neurological disorders include nusinersen for treatment of SMA, inotersen for polyneuropathy caused by hereditary transthyretin amyloidosis, and eteplirsen, golodirsen, viltolarsen, and casimersen for DMD (Table 2). A prior study administered ATL1102, a 2′-O-MOE modified gapmer designed to reduce the amount of alpha-4 integrin (VLA-4 subunit) mRNA, to 77 patients with relapsing–remitting multiple sclerosis (RRMS) and reported significant improvements in disease activity after 8 weeks of treatment [164]. Inhibition of alpha-4 integrin synthesis prevents circulating immune cells from crossing blood vessels [165]. ATL1102 was recently tested in nine non-ambulatory patients with DMD and was reported to be safe. Clinical symptoms remained stable during a 24-week treatment period [166]. Preliminary data from a recently completed phase III clinical trial showed that eplontersen, a ligand conjugated PMO-ASO inhibiting transthyretin synthesis, had an acceptable safety profile and met co-primary and secondary endpoints among patients with polyneuropathy caused by hereditary transthyretin amyloidosis [167]. Milasen, a patient-specific ASO with PS-backbone and 2′-O-MOE modifications, was administered intrathecally in a patient with a pathogenic mutation in *CLN7* causing Batten’s disease. Milasen was found to correctly restore splicing of the gene and reduce the frequency and duration of seizures in the patient [168]. Several ongoing clinical studies are assessing the safety and efficacy of other ASOs in different neurological disorders (Table 3).

Not all ASOs have shown efficacy in clinical studies. Of note, clinical trials assessing suvodirsen to skip exon 51 in DMD (NCT03907072), BIIB078 targeting C9orf72 (NCT03626012) and BIIB100 targeting XPO1 (NCT03945279) in amyotrophic lateral sclerosis, and WVE-120101 (NCT03225833) and WVE-120102 (NCT03225846) targeting mutant HTT synthesis in Huntington’s disease, failed to show efficacy. A study on ISIS-DMPKRx targeting DMPK in myotonic dystrophy was terminated due to the low concentration levels of drug in muscle (NCT02312011). The study on IONIS-DNM2-2.5 targeting dynamin 2 in centronuclear myopathies (NCT04033159) was terminated based on tolerability findings at the low-dose level. Drisapersen, which was designed to target DMD transcripts and skip exon 51 (NCT01803412), did not meet its primary end point in clinical trials and its development was discontinued.

## 7. Routes of ASO Administration in Neurological Disorders

Systemic administration of ASOs via the IV route has been commonly used in patients with neuromuscular diseases, including DMD. The subcutaneous injection of ASOs has been used in polyneuropathy caused by hereditary transthyretin amyloidosis and myotonic dystrophy. ATL1102 in RRMS and DMD has also been administered via SC injection [164,166]. Intravitreal (IVT) injection is another route for ASO administration in neuro-ophthalmologic conditions, including retinitis pigmentosa due to Usher syndrome and Leber congenital amaurosis type 10.

Transport of ASOs into the central nervous system (CNS) is challenging due to the presence of the BBB, which consists of unfenestrated endothelial or epithelial cells connected by tight junctions. The negative charge, high molecular weight, and hydrophilic nature of ASOs inhibit their diffusion across the BBB and limit the efficacy after systemic administration. Intrathecal (IT) or intraventricular injection has been used to bypass the BBB and directly deliver ASOs without delivery particles into the CNS. Several studies showed that intranasal administration is a noninvasive method to effectively deliver therapeutic agents into the CNS (Figure 6) [169,170,171]. Pathways provided by olfactory and trigeminal nerves can give access to intranasally administered drugs to enter the CNS [172]. A preclinical study demonstrated that intranasal administration of ASOs conjugated with indatraline in a mouse model of PD was able to reduce the expression of alpha-synuclein in the brain-stem monoamine nuclei [135]. Delivery particles can improve ASO transport into the CNS after systemic administration. Recent preclinical studies showed that ASOs encapsulated by glucose-coated polymeric nanocarrier [143] and peptide-conjugated ASOs [173] were able to cross the BBB and accumulate in the brain after IV injection.

## 8. Neurological Disorders with FDA-Approved ASOs

### 8.1. Spinal Muscular Atrophy

SMA is a progressive fatal neuromuscular disorder characterized by degeneration of alpha motor neurons in the anterior horn of the spinal cord that manifests as hypotonia, muscular atrophy, and muscle weakness. SMA is inherited in an autosomal recessive fashion and the incidence has been estimated to range from 1:6000 to 1:11,000 live births among the general population [174,175,176]. Oxidative stress and disturbed glutamate transport are reported to be partly involved in motor symptoms of patients with SMA [177].

The spinal motor neuron (SMN) protein is required throughout life in all species, and early embryonic lethality was reported in *SMN* knockout mice [178]. Humans have telomeric (*SMN1*) and centromeric (*SMN2*) copies of the gene and SMA develops in individuals with *SMN1* mutations. The *SMN2* gene is nearly identical to *SMN1,* but the transition of a C to a T nucleotide within exon 7 of *SMN2* creates an exonic splicing silencer that results in abnormal mRNAs lacking exon 7. *SMN2* therefore produces only about 10% of the functional protein produced by *SMN1* and cannot compensate for the low levels of SMN protein in patients with SMA. Different ASOs have been designed to block intronic splicing silencers or induce splicing enhancers in order to prevent exon 7 skipping [36,179,180].

Nusinersen (Spinraza) is a 2′-O-MOE-PS modified ASO that was approved by the FDA for marketing in 2016. It binds to a splicing silencer within intron 7 (~6000 nucleotides downstream of exon 7) and is found to restore functional mRNA. The phase III sham-controlled clinical trial in SMA patients with early infantile-onset disease (1 to 20 weeks of age) showed that nusinersen improved motor milestones in about half of individuals [9]. Furthermore, the risk of mortality or the use of permanent assisted ventilation was 47% lower in patients who received nusinersen compared to the control group. The phase III sham-controlled clinical trial in children with later onset disease (over 6 months) also reported beneficial effects of nusinersen on motor function [181]. Limited data on the efficacy of nusinersen in adult patients are available; a single-center, open-label study reported no significant objective improvements among adult cases with SMA after nusinersen treatment [182]. However, two other studies showed that nusinersen improved motor function in some adult cases [183,184].

### 8.2. Polyneuropathy Caused by Hereditary Transthyretin Amyloidosis

Hereditary transthyretin amyloidosis is a rare autosomal dominant disease characterized by extracellular accumulation of amyloid fibrils composed of the transthyretin (TTR) protein in different organs. Functional TTR protein transports retinol and thyroxine hormone. Hereditary transthyretin amyloidosis presents with several systemic features, including nephropathy, gastrointestinal, cardiovascular, neurological, and ocular manifestations. Cardiomyopathy and progressive sensorimotor neuropathy with autonomic involvement are the predominant clinical symptoms [185].

The liver is the main source of TTR production and liver transplantation has been one of the most effective treatments to reduce TTR synthesis and improve clinical symptoms in patients with hereditary transthyretin amyloidosis. Long transplant waiting times to find matched donors, high costs, and serious adverse events after the operation (including organ rejection and graft-versus-host disease) are major limitations of liver transplantation. Inotersen (Tegsedi) is a 2′-O-MOE-PS ASO that was approved by the FDA in 2018 to be administered to patients with stage 1 or stage 2 hereditary transthyretin amyloidosis. Inotersen reduces TTR synthesis by binding to the 3′ UTR of *TTR* mRNA transcripts, preventing mRNA translation. A phase III placebo-controlled clinical trial reported that over one-third of cases receiving inotersen had reduced neuropathic symptoms and about half of treated patients had an improvement in their quality-of-life scores [10]. Patisiran and vutrisiran are other FDA-approved RNA therapies to prevent TTR synthesis in individuals with hereditary transthyretin amyloidosis and were designed as small double-stranded RNAis with lipid NPs and GaINAc delivery particles, respectively [186,187].

### 8.3. Duchenne Muscular Dystrophy

DMD is a progressive X-linked recessive neuromuscular disorder that is typically caused by frameshift or nonsense variants in *DMD* that result in the production of nonfunctional and unstable dystrophin. Dystrophin with partial function can be produced when variants in *DMD* maintain the open reading frame, leading to Becker muscular dystrophy (BMD), which has milder clinical symptoms. Functional dystrophin maintains the muscle fiber membrane (sarcolemma) by linking actin filaments to proteins located in the sarcolemma. Studies have reported that DMD affects 1:5000 to 1:6000 live male births and is very rare in females [188,189]. Slow running, frequent falls, and waddling gait are early symptoms in boys with DMD from three to five years of age. Patients need mobility aids and become wheelchair-dependent by their teenage years. Most affected individuals die from respiratory/cardiac failure around 20 to 40 years of age [190,191].

ASO-mediated exon skipping has been suggested as a therapeutic agent in DMD. These ASOs prevent the splicing of an exon in pre-mRNAs to restore the reading frame and produce partially functional dystrophin, like BMD [192]. Eteplirsen (Exondys 51) was the first FDA-approved PMO-ASO, designed to skip exon 51 in patients with confirmed mutations in *DMD* that are amenable to exon 51 skipping. Eteplirsen is applicable in about 13% of patients [193]. Long-term studies with three to five years of follow-up reported that patients who received eteplirsen achieved greater walk distances during the 6-min walk test and had better pulmonary function compared to controls from a natural history cohort [7,194].

Golodirsen (Vyondys 53) and viltolarsen (Viltepso) are FDA-approved PMO-ASOs designed to induce exon 53 skipping in *DMD* pre-mRNA, and are considered therapeutic options for approximately 8% of individuals with DMD [193]. Long-term natural history data showed that these patients have a drastic decline in motor function and earlier loss of ambulation compared to other DMD subgroups [195]. Golodirsen is approved for marketing based on reported increases in dystrophin production in the skeletal muscle of patients after treatment [196]. A recent phase I/II study with 3-year follow-up reported that golodirsen was well-tolerated [11]. The study was underpowered to assess the clinical efficacy of the drug, but the loss of ambulation and decline in 6-min walk test distance were more frequent among untreated cases compared to treated patients. These findings, however, were not statistically significant [11]. An ongoing phase III placebo-controlled clinical trial is currently assessing the efficacy of golodirsen (NCT02500381).

Viltolarsen was reported to increase the levels of dystrophin in phase I/II clinical studies in a dose-dependent manner [12,197,198] and had the potential to significantly improve outcome measures including the 6-min walk test distance, time to stand from supine position, and time to complete a 10-m run/walk during 24 weeks of treatment [198]. An extension of this study showed that improvements in the time to stand from a supine position and the time to complete a 10-m run/walk were maintained for over two years [199]. An ongoing phase III placebo-controlled clinical trial is currently assessing the efficacy of viltolarsen (NCT04060199).

Casimersen (Amondys 45) is another FDA-approved PMO-ASO designed to treat the 8% of DMD patients with deletions amenable to exon 45 skipping [193]. A phase I/II clinical trial with 12 participants showed that casimersen was well tolerated [13]. The drug was approved based on data showing increased levels of dystrophin in patients after treatment [200] and an ongoing phase III clinical trial is assessing the clinical efficacy of casimersen in eligible patients (NCT02500381).

There are some concerns regarding the efficacy of the abovementioned ASOs that need to be addressed in future studies. All the approved exon-skipping ASOs for DMD produce small amounts of dystrophin, and there is relatively little evidence that the levels of dystrophin that have been measured in muscle tissue from treated patients is sufficient to alter the phenotype of the disease. Much of the data on clinical outcome measures have also utilized historical controls (instead of placebo controls), which are suboptimal comparators given shifts in the natural history of DMD over time and the inability to control for contextual effects of clinical trial participation.

## 9. White Matter Disorders

Multiple papers have reviewed ASO therapy in gray matter disorders, including Parkinson’s disease, Alzheimer’s disease, Huntington’s disease, and amyotrophic lateral sclerosis [201,202,203,204,205,206]. However, the feasibility and outcomes of ASO administration in white matter disorders have not been reviewed. The design of an ASO to inhibit the synthesis of alpha-4 integrin in MS was the first attempt to treat a white matter disorder using RNA therapy. The alpha-4 and beta-1 molecules bind together via a noncovalent bond to form the VLA-4 integrin expressed on different immune cells. The VLA-4 integrin is a cell adhesion molecule that promotes an inflammatory response by guiding immune cells to migrate across blood vessels and enter the sites of inflammation. Different pharmacological agents in MS were found to reduce the levels of alpha-4 integrin [207,208,209]. Two gapmers with modified 2′-O-MOE nucleotides have been designed to target the 3′ UTR and degrade alpha-4 mRNA by activating RNase H enzyme [164,165]. One of these ASOs, known as ATL1102, was assessed in a placebo-controlled clinical trial and showed a significant reduction in the number of new active lesions in the brain during 12 weeks of treatment [164]. Other ASOs have been designed to decrease the levels of low-affinity neurotrophin receptor and interleukin (IL)-23 in the experimental autoimmune encephalomyelitis (EAE) mouse model of MS [210,211].

Leukodystrophies (LDs) are a heterogeneous group of genetic white matter disorders. Most types of LDs are inherited in an autosomal recessive fashion but other modes of inheritance including autosomal dominant (e.g., Alexander disease or AxD) and X-linked (e.g., Pelizaeus–Merzbacher disease or PMD and X-linked adrenoleukodystrophy or X-ALD) have been found. The estimated incidence of LDs ranged from 1–2:100,000 to 1:7663 live births [212,213,214]. The discrepancies in results may be due to methodological differences including study design and sample studied. Our knowledge of LDs is limited and they are often underdiagnosed. An analysis of genetic sequencing databases showed that 1 in 4733 live births had LD-causing pathogenic variants, which is higher than prior estimations [215]. Most LDs are progressive, and treatments are often supportive. Preclinical studies demonstrate that ASO therapy is a potential treatment for AxD, Canavan disease (CD), PMD, and leukoencephalopathy with brain stem and spinal cord involvement and lactate elevation (LBSL) [37,216,217,218].

### 9.1. Alexander Disease

This progressive autosomal dominant disorder was first described over 60 years ago in an infant with intellectual disability, megalencephaly, and hydrocephalus as main clinical features [219]. AxD is a rare neurological disorder with an estimated prevalence of 1 per 2.7 million people [220]. Four AxD subtypes based on age at the disease onset include neonatal (<30 days), infantile (30 days–2 years), juvenile (2–12 years), and adult (>12 years) forms [221]. The adult form is the most frequent subtype and most affected patients present with impaired motor function, bulbar or pseudobulbar palsy, spasticity, and hyperreflexia [220,222]. Another classification scheme with two subtypes based on the age at disease onset and clinical manifestations has been proposed [223]. The mortality rate in individuals with type I AxD is two times higher than cases with type II AxD.

Although AxD is a rare condition, several studies have been focused on its underlying mechanisms and potential treatments due to the primary effects of AxD on astrocytes. The neuropathological characteristic of AxD is the cytoplasmic accumulation of glial fibrillary acidic proteins (GFAPs) that form Rosenthal fibers inside the astrocytes [224]. GFAP is an intermediate filament with predominant expression in astrocytes. Dominant gain-of-function mutations in the *GFAP* gene are the primary cause of AxD, resulting in the overproduction of GFAP with toxic function [225]. Preclinical studies showed that *GFAP* knockout mice had a mild AxD phenotype and the levels of GFAP are positively correlated with the severity of the disease [226,227,228]. Furthermore, no *GFAP*-null mutations have been found in patients. An ASO was therefore developed to suppress GFAP expression.

ASO therapy showed promising results in preclinical studies. To date, no other disease-modifying agents have been tested in AxD. Different gapmers with eight unmodified DNA nucleotides, flanked by five 2′-O-MOE modified nucleotides at each 5′ and 3′ sides, were designed and administered in mouse and rat models of AxD [216,229]. Results showed that the levels of GFAP in the CNS were reduced in a dose-dependent manner and GFAP suppression was associated with motor function improvements and reversing AxD pathology [229]. An ongoing phase I–III multicenter clinical trial will assess the safety and efficacy of ASO therapy (ION373) in 58 individuals with AxD (NCT04849741).

### 9.2. Canavan Disease

CD is a rare autosomal recessive disorder that was first described over 90 years ago in a child with progressive generalized spasticity, intellectual disability, seizures, and spongy degeneration of the subcortical white matter [230]. CD is more frequent among people of Ashkenazi Jewish descent with the estimated incidence of 1:6400 to 1:27,000 live births [231,232,233]. CD has also been reported among non-Jewish populations with high rates of consanguineous marriages [234]. Three CD subtypes based on the age at disease onset include congenital, infantile, and juvenile forms. The infantile subtype is the most common form of CD, characterized by the triad of hypotonia, poor head control, and macrocephaly [235].

Aspartoacylase (ASPA) enzyme deficiency due to mutations in the *ASPA* gene is the primary cause of CD. The main function of ASPA is to hydrolyze N-acetylaspartic acid (NAA) to produce acetate. Elevated levels of NAA can lead to disrupted osmoregulation and oxidative damage of neurons and glial cells [234,236,237,238]. Some of the current treatments in CD, including lithium citrate, lithium chloride, and sodium valproate, reduce the NAA levels in the brain [238,239,240,241]. Lithium citrate has been found to improve alertness and social interaction in patients [238]. A 16-nucleotide LNA-gapmer was recently administered in an ASPA-deficient CD mouse model via intracisternal injection and was shown to suppress the levels of neuronal NAA and reverse ataxia, Purkinje cell dendritic atrophy, and cerebellar and thalamic vacuolation [218]. Clinical trials using the LNA gapmer have not yet been initiated in CD.

### 9.3. Pelizaeus–Merzbacher Disease

PMD is the most common hypomyelinating leukodystrophy with an estimated incidence of 1–1.9:100,000 live births among males [214,242]. PMD was the first reported genetic white matter disorder after multiple members of a family showed nystagmus, developmental delay, ataxia, and spasticity, with myelin defects and diffuse sclerosis of white matter over a century ago [243,244]. The main clinical symptoms of PMD are early-onset nystagmus and hypotonia which converts to limb stiffness and spasticity later in life. Three subtypes of PMD based on the age at disease onset and clinical features include connatal (0–1 month), classic (<1 year), and transitional PMD with clinical features of both classic and connatal forms [245].

Proteolipid protein (PLP) is a transmembrane protein encoded in oligodendrocytes and is the most abundant protein in CNS myelin. Duplications, missense mutations, or deletions in *PLP1* lead to abnormal expression or absence of PLP resulting in significant myelin damage, which is the primary cause of PMD [246]. It appears that patients lacking PLP have milder PMD phenotypes and longer lifespans than patients with duplications or missense mutations in *PLP1* with nonfunctional protein. *PLP1*-knockout animal models also showed a mild phenotype [247]. Two ASOs with 2′-O-MOE modification have been designed to target either intron 5 or 3′ UTR of *PLP1* mRNA in mice and were shown to suppress the levels of PLP, in association with more than a 10-fold increase in lifespan [217]. Clinical trials have not yet been initiated.

### 9.4. Leukoencephalopathy with Brain Stem and Spinal Cord Involvement and Lactate Elevation

This is a progressive rare autosomal recessive disorder, first described in eight individuals with neurodevelopmental disabilities and characteristic features on magnetic resonance imaging, including cerebral, brain stem, cerebellar, and spinal cord involvement, about 20 years ago [248]. LBSL is characterized by progressive spastic gait, ataxia, and posterior spinal cord dysfunction. The incidence of LBSL has not been estimated, but over 100 cases have been published in the literature. A high frequency of carriers (1:95) has been reported in Finland [249].

LBSL is caused by mutations in the *DARS2* gene that encodes mitochondrial aspartyl-tRNA synthetase. This enzyme is responsible for charging aspartic acid to its cognate tRNA for protein translation. Neuronal inflammation, mitochondrial dysfunction, and apoptosis have been reported in the CNS of *DARS2*-knockout mice [250]. Most individuals with LBSL have compound heterozygote mutations consisting of one mutation in intron 2, affecting splicing of the exon 3, and one missense mutation. Full-length *DARS2* transcript is periodically produced in LBSL patients harboring the “leaky” intron 2 mutation, and it is hypothesized that correction of *DARS2* mRNA by further increasing exon 3 inclusion may restore protein function to produce a clinical benefit [251]. A prior in vitro study indicated that an ASO consisting of 2′-O-Me modified nucleotides targeting exon 3 of *DARS2* pre-mRNA and a noncomplementary tail that could induce splicing activation was able to increase levels of corrected *DARS2* mRNA including exon 3 [37]. Further studies are needed to assess the safety and efficacy of this ASO.

## 10. Neurodevelopmental Disorders

ASOs can be suitable therapeutic agents for multiple neurodevelopmental disorders (NDDs), and there are an increasing number of preclinical and clinical studies aimed at treating ultrarare NDDs and developing patient-customized therapies using ASOs. An ASO designed to treat hereditary sensory and autonomic neuropathy type IX has been successful in inducing exon 8 skipping in mutated *TECPR2* and removing the premature stop codon in cell culture [252]. Examples of patient-customized ASOs include Milasen, which was used to treat a 6-year-old girl with Batten’s disease (caused by a pathogenic mutation in *CLN7*) and Atipeksen, which was used to treat ataxia telangiectasia in a 3-year-old boy [168,253]. ASO therapy has been also tested in more common NDDs including Angelman syndrome (AS) and Dravet syndrome (DS).

### 10.1. Angelman Syndrome

AS is a severe genetic disorder that primarily affects the nervous system and was initially described about 60 years ago in three individuals with spastic gait, seizures, excessive laughter, and loss of speech [254]. AS is characterized by severe intellectual disability, developmental delay, motor dysfunction, and autism spectrum disorder. The estimated incidence is 1 in 15,000 live births [255] and is caused by the loss of the UBE3A protein, which is often due to the imprinted (silenced) paternal *UBE3A* gene and mutations affecting maternal *UBE3A* gene located on chromosome 15 (15q11.2-q13 locus). UBE3A protein is critical in brain development. Molecular studies revealed that a de novo deletion of the maternal 15q11.2-q13 locus is the most prevalent cause of AS (~70%), followed by de novo or inherited mutations within the maternal *UBE3A* (~20%), paternal uniparental disomy of chromosome 15 (~5%), and imprinting defects of maternal *UBE3A* (~5%) [256].

One promising therapeutic approach in AS is to restore *UBE3A* gene expression by introducing exogenous *UBE3A* DNA to patient cells (gene therapy) or activating the imprinted paternal *UBE3A* gene using ASO therapy. Emerging evidence has demonstrated that paternal *UBE3A* is silenced by an RNA polymerase II antisense transcript (*UBE3A-ATS*) inside the nucleus (Figure 2G) [257]. Two studies on AS mice reported that ASOs inhibiting *UBE3A-ATS* production were able to increase the levels of UBE3A protein [258,259]. One study on adult mice, however, failed to show significant improvements in most behavioral phenotypes [258]. Another study on younger mice reported partial improvements in multiple neurocognitive phenotypes [259]. The discrepancies in these results could have been due to the age of the mice at the time of treatment, as delayed AS treatment can provide limited efficacy [260,261]. Currently, there are three ongoing phase I clinical trials sponsored by Ultragenyx (NCT04259281), Hoffmann-La Roche (NCT04428281), and Ionis Pharmaceuticals (NCT05127226) to assess the safety, pharmacokinetics, and pharmacodynamics of ASOs in patients with AS.

### 10.2. Dravet Syndrome

DS is a rare developmental epileptic encephalopathy that was first introduced as severe myoclonic epilepsy in infancy (SMEI) about 45 years ago [262]. DS is characterized by convulsive seizures during the first year of life. Myoclonic, atypical absence, focal, and tonic seizures are also common among affected infants and children. Two clinical forms of DS include typical SMEI with myoclonic seizures and SMEI borderline form without the myoclonic component [263]. Developmental and psychomotor delay often develop during the second year of life. Other neurological features, including motor dysfunction, hypotonia, ataxia, and interictal myoclonus, can be observed in patients [263].

Epilepsy in patients is often resistant to treatment, and failure to fully control seizures can occur using multiple antiseizure medications. Although recent clinical studies showed promising effects of new therapeutic drugs (including stiripentol, fenfluraminein, and cannabidiol) in reducing seizure frequency [264,265,266], to date, no disease-modifying agent in DS has been approved. DS is a voltage-gated ion channelopathy and 70–80% of affected individuals have mutations in the *SCN1A* gene that suppresses the production of voltage-gated sodium channel α subunit Na_v_.1.1 [267]. A series of ASOs have been designed to target nonproductive alternative splicing sites in *SCN1A* mRNA in a DS mouse model to inhibit nonsense-mediated decay and increase the expression of productive *SCN1A* mRNA and Na_v_.1.1 (Figure 2D) [27]. One ASO was able to restore gene expression in mice after a single intracerebroventricular injection. The increase in Na_v_.1.1 after treatment was associated with significant reduction in seizure frequency and the incidence of sudden unexpected death in epilepsy. ASO therapy improved survival of the mice by over four-fold, compared to untreated DS mice [27]. An ongoing phase I/II clinical trial is assessing the safety and efficacy of increasing *SCN1A* expression in patients using ASO therapy.

Recent findings on the underlying mechanisms of DS have led to the development of other ASOs that have been successfully tested in DS mice. Studies reported that neuronal hyperexcitability due to the gain-of-function mutations in *SCN2A* encoding Na_v_.1.2 or *SCN8A* encoding Na_v_.1.6 can cause DS, and ASOs decrease seizure frequency and increase survival in DS mice by reducing the expression of either *SCN2A* or *SCN8A* mRNA [268,269].

## 11. Antisense Oligonucleotide Adverse Events

Toxicity of ASOs can be caused by: (i) type I hypersensitivity after ASO injection; (ii) binding to off-target mRNAs, which may reduce the levels of off-target proteins or alter their structure to impair function; or (iii) chemical modifications which lead to ASO-protein interaction with subsequent immune reaction and aspecific adverse events, including fever, thrombocytopenia, arthralgia, nephrotoxicity, and hepatotoxicity [270]. Reported adverse events of FDA-approved ASOs in neurological disorders are listed in Table 2. Overall, nephrotoxicity and thrombocytopenia are the most severe adverse events and the major drawbacks of ASO therapy [270]. Assessment of renal function in patients who receive ASO therapy is suggested, as the early treatment of this condition with immunosuppressive agents can improve renal function. Of 112 patients who received inotersen in a phase III clinical trial, nephrotoxicity occurred in three patients. Renal function improved in two of them after receiving glucocorticoids and cyclophosphamide, but one patient remained on hemodialysis permanently due to the late diagnosis [10]. Severe thrombocytopenia also developed in three patients treated with inotersen [10]. This condition improved in two patients after drug discontinuation and glucocorticoid administration, but one patient died due to intracerebral hemorrhage [10]. Clinical guidelines should be developed to educate physicians about ASO adverse events and how to manage these conditions.

## 12. Future Directions

RNA therapy is emerging as a potential treatment for a number of neurological disorders. The objectives of preclinical studies should therefore be adjusted to provide high validity and reliability in assessing the safety and efficacy of these drugs. Induced pluripotent stem cells (iPSCs) can be generated after reprogramming patient-derived cells and converted to CNS relevant cell types in order to evaluate ASO efficacy in an in vitro setting. Furthermore, cerebral organoids derived from iPSCs can be a useful and more complex model system to investigate neurotoxicity after treatment. Humanized animal models should be tested to assess the safety and efficacy of human-specific ASOs in an in vivo setting.

There are multiple preclinical studies reporting the therapeutic effects of ASOs with new target mRNAs in neurological conditions, including amyloid-β precursor protein in Alzheimer’s disease and Down syndrome; PMP22 in Charcot–Marie–Tooth type 1A; DUX4 overexpression in FSHD; expanded trinucleotide repeats in fragile X-associated tremor/ataxia syndrome and Huntington’s disease; HuR levels in neuropathic pain; and inhibition of inflammation in spinal cord injury and traumatic brain injury (Table 4) [74,75,87,135,271,272,273,274,275,276,277,278,279,280,281]. Further studies should assess the safety and efficacy of these ASOs.

Future studies are needed to compare the safety and efficacy of different types of oligonucleotides in neurological conditions to determine the safest and most effective RNA therapy in preclinical and clinical settings. Further delivery vehicles including nanoparticles should be developed to enhance ASO delivery in CNS disorders using less invasive routes of administration in patients. Deeper understanding of the bioavailability of each disease-specific ASO should also be evaluated in future studies.

## 13. Conclusions

ASOs are shown to be promising disease-modifying agents in several neurological disorders and are able to reduce the level of toxic proteins, increase the level of functional protein, or modify the structure of impaired protein to improve function.

## Figures and Tables

**Figure 1 pharmaceutics-14-02389-f001:**
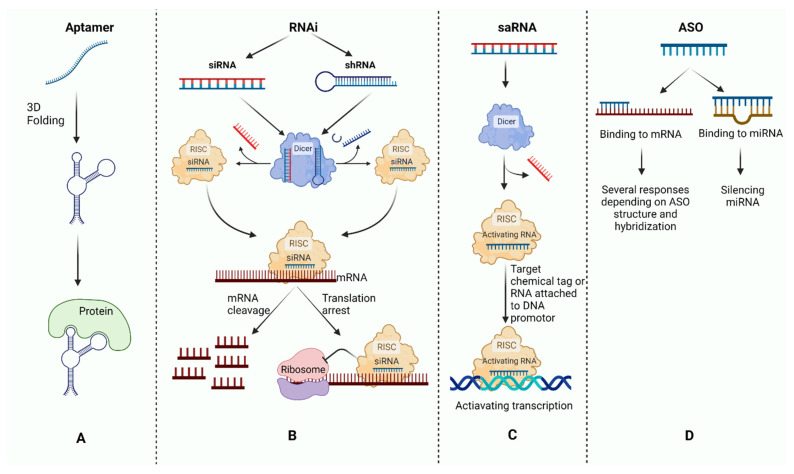
(**A**) Aptamers consist of 3D-folded single-stranded oligonucleotides with the ability to bind to proteins and peptides. They can inhibit the function of target proteins and peptides. (**B**) RNA interference (RNAi) oligonucleotides are double-stranded RNAs and include small interfering RNAs (siRNAs) and small hairpin RNAs (shRNAs). Dicer enzyme cleaves double-stranded RNAi, and antisense (guide) strands degrade target mRNAs or arrest translational processes via an RNA-induced silencing complex (RISC). RISC is a multiprotein complex that alternates gene expression. The miRNAs or siRNAs are loaded on an RISC and attach to their complementary mRNA transcripts. An RISC-associated protein or Argonaute is then activated and inhibits protein synthesis by cleaving mRNA or arresting translational process. The siRNAs transport to cell cytoplasm and are associated with Dicer/RISC, but shRNAs initially transport to cell nuclei using a vector for transcription. The product will then export to cell cytoplasm and will be associated with Dicer/RISC. (**C**) Small activating RNAs (saRNAs) are double-stranded RNAs activating gene expression via an RISC. The saRNAs are loaded on RISC and bind to chemical tags or RNA copies attached to DNA that prevent promoter activation. Argonaute is then activated and increases gene expression by removing these chemical tags and RNA copies. (**D**) Antisense oligonucleotides (ASOs) modify gene expression by binding to mRNA or noncoding RNA (miRNA). “Schematic diagram created with premade icons and templates from biorender.com (accessed on 20 August 2022)”.

**Figure 2 pharmaceutics-14-02389-f002:**
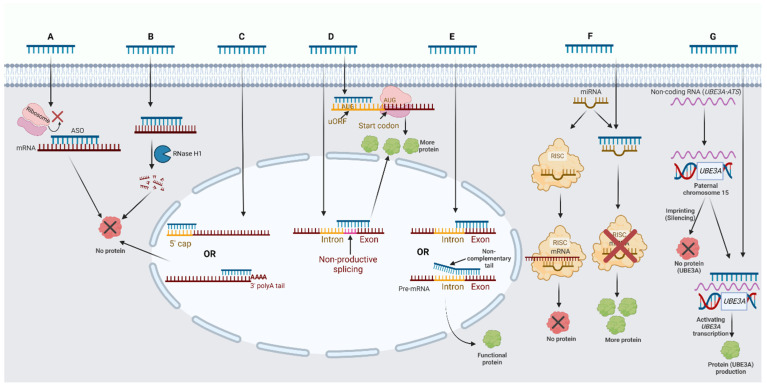
Antisense oligonucleotides (ASOs) can bind to mRNA and reduce the levels of toxic protein by (**A**) translational arrest due to the steric hindrance of ribosomal subunit binding, (**B**) inducing the RNase H1 endonuclease activity that cleaves RNA-DNA hybrids on mRNA, or (**C**) destabilizing pre-mRNA by inhibiting 5′ cap formation/modulating 3′ polyadenylation. (**D**) ASOs are able to increase gene expression by targeting upstream open reading frames within the 5′ untranslated region or nonproductive alternative splicing. (**E**) ASOs can modify mRNA splicing by targeting splicing enhancers/splicing silencers or inducing splicing enhancer by including a noncomplementary tail and alternate nonfunctional protein to improve function. (**F**) ASOs can target miRNAs and inhibit their RISC-dependent action to reduce protein translation. (**G**) Other noncoding RNAs can be the target of ASOs. ASOs targeting a noncoding transcript (*UBE3A-ATS*) were shown to activate paternal *UBE3A* gene expression in Angelman syndrome. “Schematic diagram created with premade icons and templates from biorender.com (accessed on 20 August 2022)”.

**Figure 3 pharmaceutics-14-02389-f003:**
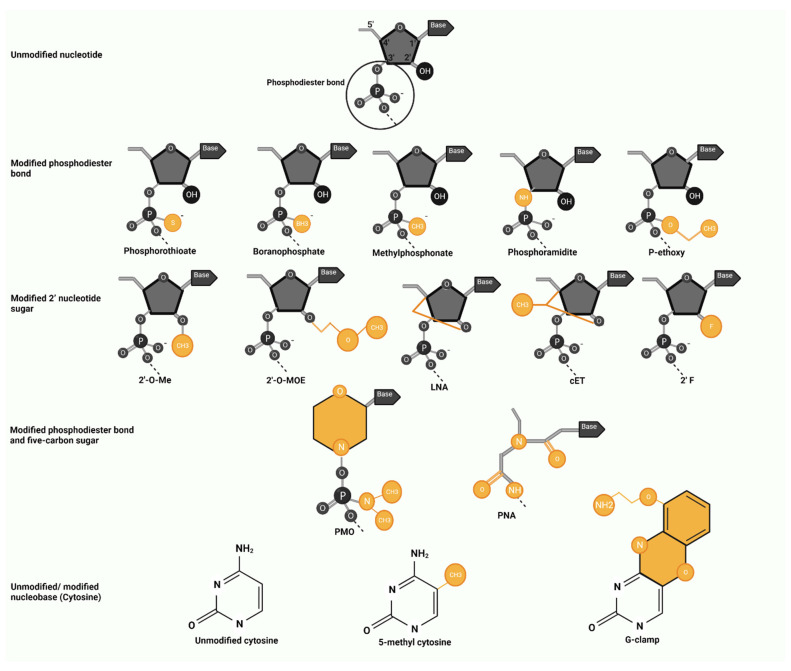
Antisense oligonucleotide modifications include modified phosphodiester bond (e.g., phosphorothioate, boranophosphate, methylphosphonate, phosphoramidite, and p-ethoxy), modified 2′ nucleotide sugar (e.g., 2′-O-methyl (2′-O-Me), 2′-O-methoxyethyl (2′-O-MOE), locked nucleic acid (LNA), constrained ethyl (cEt), and 2′-fluoro (2′ F)), modified phosphodiester bond and five-carbon sugar (e.g., phosphorodiamidate morpholino oligomer (PMO) and peptide nucleic acid (PNA)) and modified nucleobase (e.g., 5-methyl cytosine and G-clamp). “Schematic diagram created with premade icons and templates from biorender.com (accessed on 20 August 2022)”.

**Figure 4 pharmaceutics-14-02389-f004:**
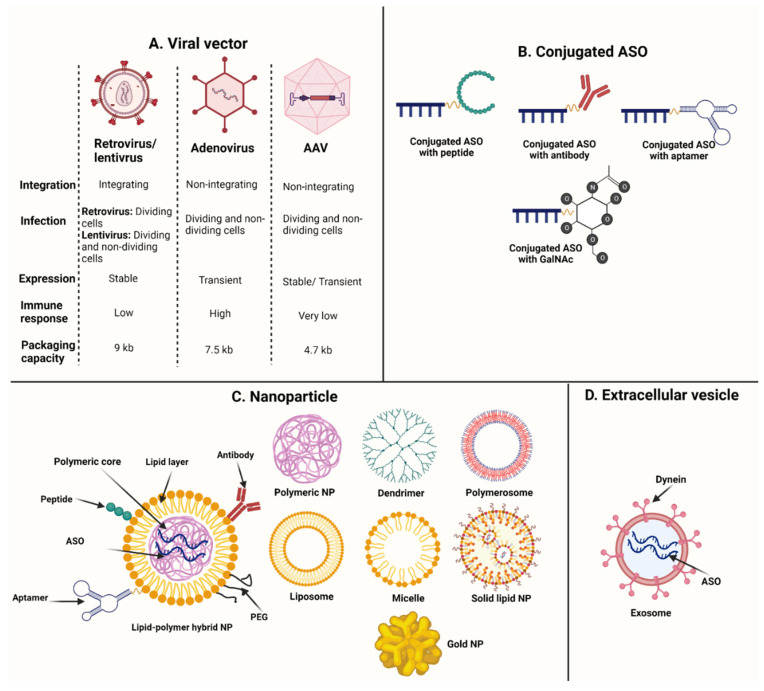
Antisense oligonucleotides can be delivered into the cells more efficiently using (**A**) viral vectors, (**B**) conjugated peptides, antibodies, and other ligands (e.g., aptamers and acetylgalactosamine (GaINAc)), (**C**) nanoparticles, or (**D**) extracellular vesicles. “Schematic diagram created with premade icons and templates from biorender.com (accessed on 20 August 2022)”.

**Figure 5 pharmaceutics-14-02389-f005:**
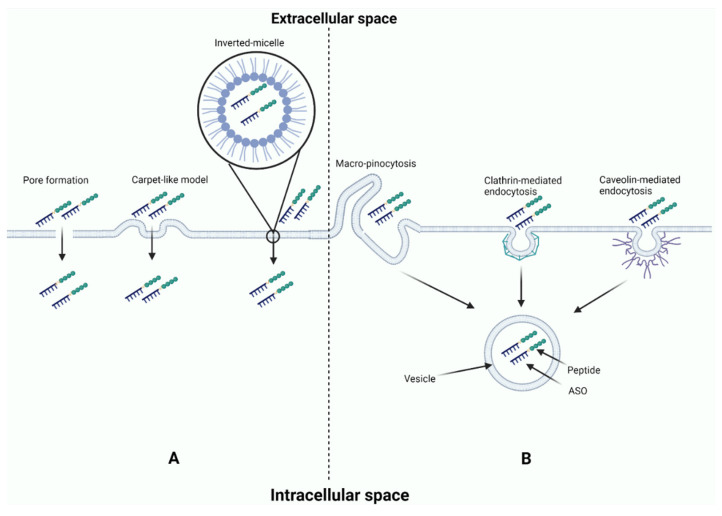
(**A**) Membrane transduction is an energy-independent pathway to deliver cell-penetrating peptides (CPPs) and their cargo into the cells via the interaction between positively charged CPPs and negatively charged phospholipids of cell membrane. This interaction forms transient structures including pores, carpet-like structures caused by destabilizing cell membrane, or inverted micelles that facilitate delivery. (**B**) Endocytic internalization is an energy-dependent pathway and is mostly directed toward macropinocytosis, clathrin-mediated, caveolin-mediated, or clathrin/caveolin-independent (not shown in this figure) pathway. “Schematic diagram created with premade icons and templates from biorender.com (accessed on 20 August 2022)”.

**Figure 6 pharmaceutics-14-02389-f006:**
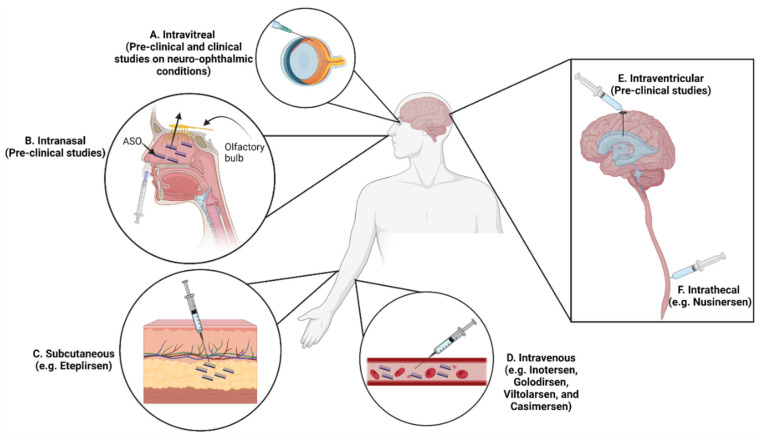
Common routes of antisense oligonucleotide (ASO) administration in neurological disorders include (**A**) intravitreal, (**B**) intranasal, (**C**) subcutaneous, (**D**) intravenous, (**E**) intraventricular, and (**F**) intrathecal injections. Among FDA-approved ASOs, eteplirsen is injected using subcutaneous route; inotersen, golodirsen, viltolarsen, and casimersen are injected using the intravenous route; and nusinersen is injected using intrathecal route. The intravitreal route of ASO delivery is currently used for preclinical studies and clinical studies on neuro-ophthalmic conditions. The intraventricular and intranasal routes of ASO administration are currently in preclinical stages. “Schematic diagram created with premade icons and templates from biorender.com (accessed on 20 August 2022)”.

**Table 1 pharmaceutics-14-02389-t001:** FDA-approved antisense oligonucleotides.

Drug (Year of Approval)	Indication	Mechanism	Design
Fomivirsen (1998)	Retinitis induced by CMV in people with AIDS [5]	Inhibitor of the major IE2 proteins synthesis in CMV (RNase H-inducing)	Phosphorothioate nucleotides 5′-G-C-G-T-T-T-G-C-T-C-T-T-C-T-T-C-T-T-G-C-G-3′
Mipomersen (2013)	Reducing LDL cholesterol concentrations in individuals with homozygous familial hypercholesterolemia [6]	Inhibitor of apolipoprotein B 100 synthesis in liver (RNase H-inducing)	2′-O-methoxyethyl and phosphorothioate nucleotides 5′-mG-mC*-mC*-mU*-mC*-A-G-T-C*-T-G-C*-T-T-C*-mG-mC*-mA-mC*-mC*-3′
Eteplirsen (2016)	Treatment of Duchenne muscular dystrophy [7]	Inducing the production of functional dystrophin through specific skipping of exon 51 in defective *DMD* variants (Splicing modulation)	phosphorodiamidate morpholino nucleotides 5′-C-T-C-C-A-A-C-A-T-C-A-A-G-G-A-A-G-A-T-G-G-C-A-T-T-T-C-T-A-G-3′
Defibrotide (2016)	Treatment of severe hepatic veno-occlusive disease [8]	It has antithrombotic, profibrinolytic, anti-ischemic, and antiadhesive features that hydrolyze clots but no specific mechanism was found	It has a natural origin achieved from polymerization of porcine intestinal mucosal DNA. It is composed of a single and double-stranded phosphodiester mixture with a length of about 50 nucleotides
Nusinersen (2016)	Treatment of spinal muscular atrophy [9]	Inducing the inclusion of exon 7 in *SMN2* gene by binding to intronic splicing silencer to produce full-length SMN protein (Splicing modulation)	2′-O-methoxyethyl and phosphorothioate nucleotides mU*-mC*-mA-mC*-mU*-mU*-mU*-mC*-mA-mU*-mA-mA-mU*-G-mC*-mU*-mG-mG
Inotersen (2018)	Treatment of hereditary transthyretin amyloidosis [10]	Inhibitor of transthyretin synthesis by binding to 3′ untranslated region of *TTR* mRNA (RNase H-inducing)	2′-O-methoxyethyl and phosphorothioate nucleotides 5′-mT-mC*-mT-mT-mG-G-T-T-A-C*-A-T-G-A-A-mA-mT-mC*-mC*-mC*
Golodirsen (2019)	Treatment of Duchenne muscular dystrophy [11]	Inducing the production of functional dystrophin through specific skipping of exon 53 in defective *DMD* variants (Splicing modulation)	phosphorodiamidate morpholino nucleotides 5′-G-T-T-G-C-C-T-C-C-G-G-T-T-C-T-G-A-A-G-G-T-G-T-T-C-3′
Viltolarsen (2020)	Treatment of Duchenne muscular dystrophy [12]	Inducing the production of functional dystrophin through specific skipping of exon 53 in defective *DMD* variants (Splicing modulation)	phosphorodiamidate morpholino nucleotides 5′-C-C-T-C-C-G-G-T-T-C-T-G-A-A-G-G-T-G-T-T-C-3′
Casimersen (2021)	Treatment of Duchenne muscular dystrophy [13]	Inducing the production of functional dystrophin through specific skipping of exon 45 in defective *DMD* variants (Splicing modulation)	phosphorodiamidate morpholino nucleotides 5′-C-A-A-T-G-C-C-A-T-C-C-T-G-G-A-G-T-T-C-C-T-G-3′

CMV: Cytomegalovirus; AIDS: Acquired immunodeficiency syndrome; LDL: Low-density lipoproteins; DNA: Deoxyribonucleic acid. (m): 2′-O-methyl or 2′-O-methoxyethyl nucleoside. (*): Methyl group at position 5 of C or U.

**Table 2 pharmaceutics-14-02389-t002:** FDA-approved antisense oligonucleotides in neurological disorders.

Drug	Disease	Target mRNA	Suggested Dose	Delivery Route	Adverse Events *
Nusinersen (Spinraza)	Spinal muscular atrophy	*SMN2* to increase exon 7 inclusion	12 mg Treatment includes four loading doses. The first three doses are injected at 14-day intervals and the fourth dose should be injected 30 days after the third dose. The drug is then administered every four months.	IT	Nephrotoxicity, coagulation defects, thrombocytopenia, and IT-related adverse events including headaches, fever, and meningitis
Inotersen (Tegsedi)	Polyneuropathy caused by hereditary transthyretin amyloidosis	*TTR* to prevent protein synthesis	300 mg Once per week	SC	Injection site reaction, headaches, fever, nausea, decreased appetite, liver inflammation, and nephrotoxicity are common adverse events. One patient who received inotersen in a clinical trial died after intracranial hemorrhage due to thrombocytopenia
Eteplirsen (Exondys 51)	Duchenne muscular dystrophy	*DMD* to skip exon 51	30 mg/kg Once per week as 35- to 60-min infusion	IV	Hypersensitivity reaction (e.g., bronchospasm, chest pain, cough, tachycardia, and urticaria), headaches, nausea, vomiting, upper respiratory tract infection, nasopharyngitis, balance disorder, contact dermatitis, and arthralgia
Golodirsen (Vyondys 53)	Duchenne muscular dystrophy	*DMD* to skip exon 53	30 mg/kg Once per week as 35- to 60-min infusion	IV	Hypersensitivity reaction, headaches, nausea, vomiting, cough, fever, abdominal pain, nasopharyngitis, falls, and nephrotoxicity
Viltolarsen (Viltepso)	Duchenne muscular dystrophy	*DMD* to skip exon 53	80 mg/kg Once per week as 60-min infusion	IV	Upper respiratory tract infection, injection site reaction, cough, fever, contusion, arthralgia, diarrhea, nausea, vomiting, abdominal pain, reduced ejection fraction, urticarial, and nephrotoxicity
Casimersen (Amondys 45)	Duchenne muscular dystrophy	*DMD* to skip exon 45	30 mg/kg Once per week as 35- to 60-min infusion	IV	Upper respiratory tract infection, cough, fever, headaches, arthralgia, pain in mouth and throat, and nephrotoxicity

* It has been recommended to monitor renal function in patients who receive ASO therapy before and during treatment. IT: Intrathecal; SC: Subcutaneous; IV: Intravenous.

**Table 3 pharmaceutics-14-02389-t003:** Ongoing active clinical trials using antisense oligonucleotides in neurological disorders based on clinicaltrials.gov (accessed on 28 September 2022) *.

Identifier	Disease	Drug	Phase	Delivery Route	Notes
NCT04849741	Alexander disease	ASO inhibiting GFAP synthesis (ION373)	I–III	IT	The drug or placebo will be injected for a 60-week period. All participants will then receive ION373 for a 60-week open-label treatment period.
NCT03186989	Alzheimer’s disease and frontotemporal dementia	ASO inhibiting tau protein synthesis (IONIS-MAPTRx)	I/II	IT	The study will include two parts. In part 1, the ASO or placebo will be administered at 4-week intervals over the course of a 13-week treatment period for four dose levels. In part 2, ASO will be administered at quarterly intervals for 48 weeks.
NCT04931862	Amyotrophic lateral sclerosis or frontotemporal dementia with documented mutation of G4C2 repeat expansion in the first intronic region of the *C9orf72* gene	ASO targeting *C9orf72* mRNA to selectively reduce *C9orf72*-repeat-containing transcripts (WVE-004)	I/II	IT	Different doses of WVE-004 or artificial cerebrospinal fluid (placebo) will be administered
NCT04768972	Amyotrophic lateral sclerosis with *FUS* mutation	ASO inhibiting FUS protein synthesis (ION363)	III	IT	The study will include two parts. Part 1 will consist of participants that will be randomized in a 2:1 ratio to receive a multidose regimen of ION363 or placebo for a period of 61 weeks, followed by Part 2, in which participants will receive ION363 for a period of 85 weeks.
NCT04494256	Amyotrophic lateral sclerosis with no mutation in *SOD1* or *FUS* gene	ASO inhibiting ataxin-2 synthesis (BIIB105)	I	IT	The study will include three loading doses for three days, followed by two or five maintenance doses.
NCT04856982	Amyotrophic lateral sclerosis with *SOD1* mutation	ASO inhibiting SOD1 protein synthesis (Tofersen)	III	IT	The drug will be administered on days 1, 15, 29, and every 28 days thereafter for up to 2 years. In the next part, participants who received placebo before, will receive tofersen 100 mg on days 1, 15, 29, and every 28 days thereafter for up to 2 years. Participants who received tofersen before, will receive tofersen 100 mg on days 1, 29, and every 28 days thereafter for up to 2 years, with a dose of placebo on day 15 to maintain the study blind
NCT04259281	Angelman Syndrome	ASO inhibiting *UBE3A* antisense transcript to increase paternal *UBE3A* expression (GTX-102)	I/II	IT	Treatment will include 2 mg, 3.3 mg, or 5 mg for 3–4 monthly doses followed by a quarterly maintenance regimen
NCT04428281	Angelman Syndrome	ASO inhibiting *UBE3A* antisense transcript to increase paternal *UBE3A* expression (RO7248824)	I	IT	The drug will be administered in different dose levels over a period of 8 weeks, with a minimum of approximately 4 weeks between each dose administration. In the long-term extension part RO7248824 will be administered up to 10 doses in selected dose levels with a minimum of approximately 16 weeks between each dose administration.
NCT05127226	Angelman Syndrome	ASO inhibiting *UBE3A* antisense transcript to increase paternal *UBE3A* expression (ION582)	I/II	IT	ION582 will be administered in different doses over a period of 13 weeks, with a minimum of approximately 4 weeks between each dose administration.
NCT04123626	Autosomal dominant retinitis pigmentosa due to the P23H mutation in the *RHO* gene	ASO inhibiting P23H protein expression while preserving expression of the wild type rhodopsin protein (QR-1123)	I/II	IVT	The study will comprise up to 8 single dose and repeat dose cohorts. In the single dose, patients will receive a single injection. In the repeat dose cohorts, subjects will be randomized to receive either a unilateral injection of every 3 months or a unilateral sham-procedure every 3 months
NCT04442295	Dravet syndrome	ASO targeting nonproductive alternative splicing in *SCN1A* mRNA to upregulate Na_v_1.1 (STK001)	I/II	IT	In one group single ascending doses including 10 mg, 20 mg, 30 mg, and 45 mg will be administered, and in another group, multiple ascending doses including 20 mg, 30 mg, and 45 mg will be administered
NCT04906460	Duchenne muscular dystrophy	Exon-skipping ASO to promote skipping over exon 53 in *DMD* pre-mRNA (WVE-N531)	I/II	IV	Up to 4 dose levels of WVE-N531 will be administered at least 4 weeks apart
NCT04004065	Duchenne muscular dystrophy	Peptide-conjugated eteplirsen to promote skipping over exon 51 in *DMD* pre-mRNA (vesleteplirsen or SRP-5051)	II	IV	Participants will receive escalating dose levels of treatment, every 4 weeks, for up to 75 weeks during Part A of the study. Once the doses have been selected for Part B, all participants who have completed Part A will transition to Part B. In this part participants will receive treatment at the doses selected based on data from Part A every 4 weeks, for up to 2 years.
NCT05524883	Duchenne muscular dystrophy	Exon-skipping ASO to promote skipping over exon 51 in *DMD* pre-mRNA (DYNE-251)	I/II	IV	DYNE-251 or placebo will be administered 6 times over 24 weeks. Then, DYNE-251 will be administered up to 30 times over 120 weeks in all participants
NCT05032196	Early Huntington’s disease	Allele selective ASO targeting single nucleotide polymorphism (SNP)-3 to reduce mutant HTT synthesis (WVE-003)	I/II	IT	Injection of 4 doses of WVE-003 or placebo to find out the safety and pharmacokinetics of the drug
NCT03761849	Huntington’s disease	Allele nonselective ASO lowering mutant and wild-type HTT synthesis (RO7234292)	III	IT	The drug will be administered every 8 or 16 weeks to compare its safety and efficacy to placebo that will be injected every 8 weeks
NCT04855045	Leber congenital amaurosis 10 due to the c.2991+1655A>G	ASO targeting mutated *CEP290* pre-mRNA to redirect normal splicing (Sepofarsen)	II/III	IVT	The study will consist of two parts: an open-label dose escalation part with three planned dose groups, followed by a double-masked randomized part. Patients will receive an initial loading dose, followed by maintenance doses every 6 months
NCT03913143	Leber congenital amaurosis 10 due to the c.2991+1655A>G	ASO targeting mutated *CEP290* pre-mRNA to redirect normal splicing (Sepofarsen)	II/III	IVT	Treatment will include an initial loading dose, followed by maintenance doses at month 3 and every 6 months
NCT04165486	Multiple system atrophy	ASO inhibiting alpha-synuclein synthesis (ION464)	I	IT	The study will include two parts. Part 1 of the study will consist a screening period of up to 6 weeks, a treatment period of 12 weeks, and a follow-up period of 24 weeks. The study duration for each participant in part 2 will be approximately 96 weeks, which will consist a 72-week treatment period and a 24-week follow-up period.
NCT05481879	Myotonic dystrophy type I	ASO inhibiting DM1 protein kinase synthesis (DYNE-101)	I/II	IV	The study will include screening period (up to 6 weeks), a multiple-ascending dose placebo-controlled period (24 weeks), a treatment period (24 weeks) and a long-term extension period (96 weeks).
NCT04485949	Newly diagnosed glioblastoma multiforme	IGV-001, containing autologous glioblastoma cells treated with antisense oligonucleotide (IMV-001) targeting IGF-1R, encapsulated in biodiffusion chambers; followed by radiotherapy and temozolomide	II	Abdominal rectus sheath	Biodiffusion chambers including IGV-001 or placebo will be implanted and then, remove at approximately 48 h after implantation. After 6 weeks of ASO therapy, participants will receive radiotherapy per institutional standards for 5 days per week along with temozolomide 75 mg/m^2^ orally, once daily for up to 12 weeks followed by temozolomide 150 to 200 mg/m^2^, orally, on Days 1 to 5 of each 28-day cycle for up to 6 cycles
NCT03976349	Parkinson’s disease	ASO inhibiting LRRK2 synthesis (ION859)	I	IT	Study will have two parts. In part A, Participants will receive a single injection in different doses, and in part B, Participants will receive a single injection in different doses on multiple days
NCT04539041	Progressive supranuclear palsy	ASO targeting *MPAT* mRNA to reduce tau expression (NIO752)	I	IT	Drug in different doses or placebo will be administered in 6 cohorts over 3 months
NCT05160558	Spinocerebellar Ataxia 3	ASO inhibiting ATXN3 protein (BIIB132)	I	IT	The study will consist of five cohorts to administer drug or placebo every four weeks for up to 85 days
NCT05085964	*USH2A*-associated retinitis pigmentosa	ASO targeting *USH2A* mRNA to skip exon 13 (Ultevursen or QR-421a)	II	IVT	This is an extension of a completed phase I/II trial (Stellar study with NCT03780257 identifier). The QR-421a Injection will be in one or both eyes of all participants and repeat every 6 months for an anticipated period of 24 months (Helia study)
NCT05176717	*USH2A*-associated retinitis pigmentosa	ASO targeting *USH2A* mRNA to skip exon 13 (Ultevursen or QR-421a)	II/III	IVT	Patients with early to moderate vision loss will receive loading dose of 180 µg (group 1) or 60 µg (group 2) and maintenance dose of 60 µg. The third group receives sham-procedure (Celeste study)
NCT05158296	*USH2A*-associated retinitis pigmentosa	ASO targeting *USH2A* mRNA to skip exon 13 (Ultevursen or QR-421a)	II/III	IVT	Patients with advanced vision loss will receive loading dose of 180 µg (group 1) or 60 µg (group 2) and maintenance dose of 60 µg. The third group receives sham-procedure (Sirius study)

* A full search of clinicaltrials.gov up to September 2022 was performed to identify any active ongoing clinical trial using antisense oligonucleotides in neurological disorders. Reports of all countries were included. Clinical trials with “Terminated,” “Complete,” or “Unknown” status were excluded. IVT: Intravitreal; IV: Intravenous; IT: Intrathecal; SC: Subcutaneous.

**Table 4 pharmaceutics-14-02389-t004:** Preclinical studies reporting the therapeutic effects of antisense oligonucleotides with new target mRNAs in neurological conditions.

Authors	Disease	Mechanism	Study Setting (Route of Administration/Type of Animals or Cells)
Kumar et al. [271]; Poon et al. [272]; Chang et al. [273]	Alzheimer’s disease	Reducing the levels of toxic amyloid-β precursor protein	In vivo ICV in mice
Zhao et al. [274]	Charcot–Marie–Tooth type 1A	Reducing the levels of *PMP22* mRNA	In vivo SC in mice
Chang et al. [273]	Down syndrome	Reducing the levels of toxic amyloid-β precursor protein	In vitro Patient-derived fibroblasts
Ansseau et al. [275]; Chen et al. [276]; Lim et al. [75]	Facioscapulohumeral muscular dystrophy (FSHD)	Reducing the levels of *DUX4* mRNA	Ansseau et al. and Lim et al.: In vivo intramuscular injection in mice Chen et al.: In vitro patient-derived myotubes In vivo Electroporation of ASO into FSHD patient muscle xenografts in mice
Derbis et al. [277]	Fragile X-associated tremor/ataxia syndrome	Targeting the expanded CGG repeat in *FMR1* mRNA	In vitro Patient-derived fibroblasts In vivo ICV in mice
Gagnon et al. [74]	Huntington’s disease	Targeting the expanded CAG repeat in *HTT* mRNA	In vitro Patient-derived fibroblasts
Borgonetti and Galeotti [278]	Neuropathic pain	Reducing the levels of *HuR* mRNA	In vivo Intranasal or IT injection in mice
Alarcón-Arís et al. [135]	Parkinson’s disease	Reducing the levels of alpha-synuclein	In vivo Intranasal in mice
Pelisch et al. [87]	Spinal cord injury	Reducing the levels of CCL3 chemokine to inhibit inflammation	In vitro Bone marrow-derived macrophages of mice In vivo IT in mice
Shohami et al. [279]; Ghirinkar et al. [280]; Lu et al. [281]	Traumatic brain injury	Shohami et al.: Reducing the levels of acetylcholinesterase Ghirinkar et al.: Reducing the levels of monocyte chemoattractant protein 1 Lu et al.: Reducing the levels of transient receptor potential vanilloid type 4	In vivo Shohami et al.: ICV in mice Ghirinkar et al.: Infusion into the injured area in brain of rats Lu et al.: ICV in rats

ICV: intracerebroventricular; IT: intrathecal.

## Data Availability

Not applicable.

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
