# Peer review of "Antisense Oligonucleotide Therapy for the Nervous System: From Bench to Bedside with Emphasis on Pediatric Neurology"

_pharmaceutics, 2022, doi:10.3390/pharmaceutics14112389_

Round 1
Reviewer 1 Report
Dear authors,
After the review report, I have several comments: you should mention how was realized the figures, in the legend; you should present comments related to the therapeutic applications of biomolecules in neurodegenerative diseases; the bioavailability role should be extended at other problems that interact with neurological problems - oxidative stress.
Best regards!
Author Response
We sincerely thank the editors and reviewers for providing us with their insightful comments on our manuscript pharmaceutics-1975540. The paper has been revised extensively based on these comments. Most importantly we have changed the name of the paper to “Antisense Oligonucleotide Therapy for the Nervous System: From Bench to Bedside with Emphasis on Pediatric Neurology” to highlight our focus on pediatric neurological conditions while we do also discuss adult diseases. Consequently, we have added new sections to the manuscript and a new table. We now have added a section discussing two neurodevelopmental disorders, namely Angelman syndrome and Dravet Syndrome. All tables and figures are placed at the end of manuscript. Below, we provide a point-by-point response to the reviewers’ comments and enclose a tracked changes and clean version of the revised manuscript.
Reviewer 1: “After the review report, I have several comments: you should mention how was realized the figures, in the legend; you should present comments related to the therapeutic applications of biomolecules in neurodegenerative diseases; the bioavailability role should be extended at other problems that interact with neurological problems - oxidative stress.”
Response: We appreciate these comments. The legends have been revised to better explain the figures.
One of our main goals for this paper was to explain the feasibility and outcomes of ASO therapy in disorders that have never been reviewed, and the role of biomolecules in neurodegenerative disease have been extensively reviewed elsewhere. Importantly, based on this comment and comments further below, we have opted to focus primarily our paper on pediatric neurological disorders. We have therefore changed the title of the manuscript to reflect the age focus. In addition, we have added a table (Table 4) to discuss more about other neurological conditions with associated citations which addresses both adult and pediatric neurological disorders.
We have added information about the role of oxidative stress in motor dysfunction of patients with SMA. However, we are concerned that further lengthening the manuscript to include the bioavailability role in diseases that are not primary neurological disorders would expand the scope of this review too far beyond its original intent.
Reviewer 2 Report
The authors reviewed the therapeutic use of antisense oligo nucleotides (ASOs), particularly in neurological disorders. On the whole the review has provided reasonable coverage and some depth, but there are issues that need to be addressed.
Major issues:
1. Clathrin-dependent endocytosis typically occur with receptor-mediated endocytic uptake. Would this be an important mechanism for the uptake of CPPs which are non-receptor mediated?
2. Exosomes are but one type of the heterogeneous extracellular vesicles (ECs), but only exosomes are mentioned. In any case, could ASOs be readily introduced into exosomes or other ECs like the endogenous RNAs and DNAs? Co-incubation means that the ASOs are delivered into target cells by adhering to the surface of exosomes. Would this mode of surface clinging hitchhiking be useful in efficient delivery?
3. Missing from Table 2 are the neurological disorders and their corresponding gene/variants targeted by the FDA approved ASO therapeutics. A brief annotation would save the reader from having to flip back to Table 1.
4. The schematic diagram in Fig 6 is not useful because the different routes of administration is not accompanied by information on the type of ASO that would be better suited for a particular route, or which nature is such that their delivery would necessitate direct CNS administration. This figure needs to be improved.
5. Following immediately after section 8 on “Neurological disorders with FDA-approved ASOs” is section 9 simply entitled “White matter disorders”. The transition is abrupt and there is no explanation as to why the authors focused on these class of neurological disorders, or why are these more amenable to targeting by ASOs. The lead in should be better elaborated.
6. While the authors’ focus on approved ASOs and those in clinical trials is fine, interesting experimental ASO therapies with key neurological disease that has shown effect in animals should also be mentioned (eg., targeting of APP for Alzheimer’s and Downs Syndrome, and α-synuclein for Parkinson’s). Also, what about experimental ASO’s targeting inflammatory factors to alleviate allodynia after spinal cord injury?
7. ASO obviously have its disadvantages but the authors did not highlight these. The latter would be important for a balance review. Why, for example, is nephrotoxicity a common drawback of ASO-based therapy? These and other points should be appropriately elaborated.
8. While the focus of the review is on ASOs, the authors should nonetheless compare and contrast these with other RNA based therapies, such as RNAi-based strategies, in terms of suitability and efficacy for a given neurological disorder (for example in the targeting of diseases-specific allelic expressions and splice variants).
Other issues:
1. The authors stated in the schematic diagrams Figs 1-6 “Created with biorender.com”. I think the latter refers to a website domain name. It would probably be more accurate to state “Schematic diagram created with pre-made icons and templates from biorender.com”.
2. The legend of Table 3 states: ” Ongoing active clinical trials using antisense oligonucleotides in neurological disorders based on clinicaltrials.gov”. The search strategy used by the authors to gather these trials listed should be better described.
3. There are typographical and language errors. For example, legend for Fig 5B should begin with “Endocytic…”
Author Response
We sincerely thank the editors and reviewers for providing us with their insightful comments on our manuscript pharmaceutics-1975540. The paper has been revised extensively based on these comments. Most importantly we have changed the name of the paper to “Antisense Oligonucleotide Therapy for the Nervous System: From Bench to Bedside with Emphasis on Pediatric Neurology” to highlight our focus on pediatric neurological conditions while we do also discuss adult diseases. Consequently, we have added new sections to the manuscript and a new table. We now have added a section discussing two neurodevelopmental disorders, namely Angelman syndrome and Dravet Syndrome. All tables and figures are placed at the end of manuscript. Below, we provide a point-by-point response to the reviewers’ comments and enclose a tracked changes and clean version of the revised manuscript.
- “Clathrin-dependent endocytosis typically occur with receptor-mediated endocytic uptake. Would this be an important mechanism for the uptake of CPPs which are non-receptor mediated?”
Response: Clathrin-mediated endocytosis has been reported as an important mechanism for the uptake of common CPPs. As you’ve stated, clathrin-dependent endocytosis is a receptor-mediated endocytic uptake and peptides in CPPs need to bind to receptors in this pathway. However, these receptors are mostly unknown. One study showed that syndecan-4, one of the heparin sulfate proteoglycans is a receptor for clathrin-mediated endocytosis of arginine-rich CPPs (Kawaguchi et al. Syndecan-4 is a receptor for clathrin-mediated endocytosis of arginine-rich cell-penetrating peptides. Bioconjugate chemistry. 2016;27(4):1119-30). Citations for endocytic pathways related to CPP delivery are now included in the revised version.
- “Exosomes are but one type of the heterogeneous extracellular vesicles (ECs), but only exosomes are mentioned. In any case, could ASOs be readily introduced into exosomes or other ECs like the endogenous RNAs and DNAs? Co-incubation means that the ASOs are delivered into target cells by adhering to the surface of exosomes. Would this mode of surface clinging hitchhiking be useful in efficient delivery?”
Response: We thank the reviewer for this question. We have added a new section for ASO delivery in the revised version to explain the role of extracellular vesicles. Most studies on ASO delivery with these particles have focused on the safety and efficacy of exosomes and evidence on other vesicles is limited. We believe that simple co-incubation can improve ASO delivery significantly. A recent in-vivo study on the DMD mouse model showed that exosomes loaded with exon-skipping ASOs using co-incubation increased the levels of dystrophin by 18 fold in DMD mouse model compared to naked ASOs (Gao et al. Anchor peptide captures, targets, and loads exosomes of diverse origins for diagnostics and therapy. Science Translational Medicine. 2018;10(444):eaat0195). However, please note that oligonucleotides get inside the exosomes upon co-incubation (Didiot MC et al. Exosome-mediated delivery of hydrophobically modified siRNA for huntingtin mRNA silencing. Molecular Therapy. 2016;24(10):1836-47). We believe that ASO delivery cannot be improved if they weakly adhere to the surface of exosomes.
- 3. “Missing from Table 2 are the neurological disorders and their corresponding gene/variants targeted by the FDA approved ASO therapeutics. A brief annotation would save the reader from having to flip back to Table 1.”
Response: Two columns are added to Table 2 to describe the disease and target mRNAs of the FDA-approved ASOs.
- “The schematic diagram in Fig 6 is not useful because the different routes of administration is not accompanied by information on the type of ASO that would be better suited for a particular route, or which nature is such that their delivery would necessitate direct CNS administration. This figure needs to be improved.”
Response: We added the delivery routes of the FDA-approved ASOs (SC, IV, and IT) in neurological disorders and the current state of using other routes for ASO delivery to make the figure more informative. The intent behind Figure 6 was to visualize all the possible routes for ASO delivery in neurological disorders. However, there is insufficient data to determine if a chemical modification is superior to others for a particular route. For instance, FDA-approved ASOs with IT injection have 2’-O-MOE modification but no data showed that this modification is better suited for IT injection compared to PMO-ASOs or other modifications.
- 5. “Following immediately after section 8 on “Neurological disorders with FDA-approved ASOs” is section 9 simply entitled “White matter disorders”. The transition is abrupt and there is no explanation as to why the authors focused on these class of neurological disorders, or why are these more amenable to targeting by ASOs. The lead in should be better elaborated.”
Response: We have changed the title of the manuscript to states specifically our focus on pediatric neurological disorders. Within the pediatric field genetic disorders of myelination are becoming a major target for ASO therapy. In the revised version, we added an introductory paragraph in the white matter disorder section that discusses why we are including this category.
- “While the authors’ focus on approved ASOs and those in clinical trials is fine, interesting experimental ASO therapies with key neurological disease that has shown effect in animals should also be mentioned (eg., targeting of APP for Alzheimer’s and Downs Syndrome, and α-synuclein for Parkinson’s). Also, what about experimental ASO’s targeting inflammatory factors to alleviate allodynia after spinal cord injury?”
Response: Based on your comments we have added Table 4 while focusing still on pediatric neurological disorders. In the manuscript, we also added a paragraph to further address your comment.
- 7. “ASO obviously have its disadvantages but the authors did not highlight these. The latter would be important for a balance review. Why, for example, is nephrotoxicity a common drawback of ASO-based therapy? These and other points should be appropriately elaborated.”
Response: We appreciate the guidance here and have now added a new section titled “Antisense oligonucleotide adverse events” to address this issue. Also, Table 2 includes a column listing reported adverse events of each FDA-approved ASO in neurological conditions.
- 8. “While the focus of the review is on ASOs, the authors should nonetheless compare and contrast these with other RNA based therapies, such as RNAi-based strategies, in terms of suitability and efficacy for a given neurological disorder (for example in the targeting of diseases-specific allelic expressions and splice variants).”
Response: A paragraph is added to the introduction of paper to discuss about other RNA therapeutic agents. Please consider that evidence is lacking to compare the safety and efficacy of different RNA therapeutic agents.
Reviewer 2 Other Comments:
- “The authors stated in the schematic diagrams Figs 1-6 “Created with biorender.com”. I think the latter refers to a website domain name. It would probably be more accurate to state “Schematic diagram created with pre-made icons and templates from biorender.com”.”
Response: We replaced “Created with biorender.com” to “Schematic diagram created with pre-made icons and templates from biorender.com”.
- “The legend of Table 3 states: ” Ongoing active clinical trials using antisense oligonucleotides in neurological disorders based on clinicaltrials.gov”. The search strategy used by the authors to gather these trials listed should be better described.
Response: We added a footnote to Table 3 explaining the search strategy.
- There are typographical and language errors. For example, legend for Fig 5B should begin with “Endocytic…”
Thank you again for these insightful comments. We believe the revised manuscript is improved by using your comments. We have reviewed the manuscript for typographical and language errors for the revised version.
I hope that this current version is now acceptable for publication.
Round 2
Reviewer 1 Report
Dear authors,
My comments related to bioavailability should be added as a future valorization of the paper.
Best regards!
Author Response
We sincerely thank the reviewer for providing us with the insightful comment on our manuscript pharmaceutics-1975540 entitled “Antisense Oligonucleotide Therapy for the Nervous System: From Bench to Bedside with Emphasis on Pediatric Neurology”. Below, we provide a point-by-point response to the reviewer’s comment and enclose a tracked changes of the revised manuscript.
“My comments related to bioavailability should be added as a future valorization of the paper.”
Response: We appreciate the comment. In the first revision, you commented: “the bioavailability role should be extended at other problems that interact with neurological problems - oxidative stress.” We’re not certain if we understand what you mean in this comment. We add a sentence in “Future direction” to clarify that deeper understanding of the bioavailability of each disease-specific ASO should be assessed in future studies. We hope that this revision addresses your comment properly and this current version is now acceptable for publication
Thanks,
Ali Fatemi
Corresponding Author
Reviewer 2 Report
The authors have made substantial revisions to their manuscript, which enhanced its content and clarity, with my previous comments addressed.
Author Response
Thanks so much and we hope you enjoyed reading the paper.
Thanks,
Ali Fatemi
Corresponding Author